# BIN1 regulates actin-membrane interactions during IRSp53-dependent filopodia formation
Laura Picas [1] ✉, Charlotte André-Arpin[1], Franck Comunale[2], Hugo Bousquet[3], Feng-Ching Tsai [4], Félix Rico [5], Paolo Maiuri [6], Julien Pernier[7], Stéphane Bodin[2], Anne-Sophie Nicot[8], Jocelyn Laporte [9], Patricia Bassereau [4], Bruno Goud [3], Cécile Gauthier-Rouvière [2,10] ✉ & Stéphanie Miserey [3,10] ✉

Amphiphysin 2 (BIN1) is a membrane and actin remodeling protein mutated in congenital and adult centronuclear myopathies. Here, we report an unexpected function of this N-BAR domain protein BIN1 in filopodia formation. We demonstrated that BIN1 expression is necessary and sufficient to induce filopodia formation. BIN1 is present at the base of forming filopodia and all along filopodia, where it colocalizes with F-actin. We identify that BIN1-mediated filopodia formation requires IRSp53, which allows its localization at negatively-curved membrane topologies. Our results show that BIN1 bundles actin in vitro. Finally, we identify that BIN1 regulates the membrane-to-cortex architecture and functions as a molecular platform to recruit actin-binding proteins, dynamin and ezrin, to promote filopodia formation.

Cellular morphologies powered by the actin cytoskeleton are essential to support developmental processes and cellular functions. Among the actin-driven structures produced by cells, filopodia assist relevant processes such as axonal guidance, establishing neuronal synapses, zippering of epithelial sheets, or myoblast fusion[1–3]. Filopodia are specialized 60–200 nm diameter finger-like structures made of paired actin filaments of a few to hundred microns in length[4,5]. The remodeling of the membrane/actin interface plays a central role in the formation and shape maintenance of filopodia. This function is driven by modular proteins that connect actin microfilaments with the plasma membrane, typically via phosphatidylinositol 4,5-bisphosphate $(PI(4,5)P_2)$-rich interfaces, as in the case of ezrin-radixin-moesin (ERM) domain proteins[6–8], or the inverse-Bin1-Amphiphysin-Rvs (I-BAR) domain protein IRSp53[9,10].

Bin/Amphiphysin/Rvs (BAR) domain proteins are multi-functional effectors characterized by a modular architecture consisting of an intrinsically curved membrane-binding BAR domain followed by auxiliary domains mediating protein-protein interactions and Rho GTPase signaling domains[11,12]. One of the most prevalent auxiliary domains is the Src homology 3 (SH3)[13], and in many BAR domain proteins, it binds directly with cytoskeletal assembly factors and the dynamin GTPase[14–17]. Despite the role of dynamin in endocytosis[18], recent works support its central function as a multi-filament actin-bundling protein that propels cellular protrusions during myoblast fusion[19,20]. Indeed, dense F-actin structures at the myoblast fusion site facilitate the formation of protrusions, allowing cell membrane juxtaposition that powers the mechanical forces required for cell-cell fusion and undergo myoblast differentiation[21,22]. However, what precisely controls dynamin recruitment to initiate the formation of actin-rich protrusions at the plasma membrane remain unclear.

In skeletal muscle cells, the N-BAR domain protein BIN1/Amphiphysin 2 is a major binding partner of dynamin[23,24], and its expression appears highly induced during skeletal muscle differentiation[25,26]. Indeed, the perturbed BIN1-dynamin interaction is the cause of centronuclear myopathies, a heterogeneous group of inherited muscular disorders characterized by fiber atrophy and muscle weakness[24,27]. Interestingly, the muscle-specific BIN1 isoform (BIN1 isoform 8), in contrast to the other BIN1/amphiphysin isoforms, displays a phosphoinositide (PI)-binding motif responsible for its targeting to the plasma membrane, mainly by interacting with $PI(4,5)P_2$[26]. We previously showed that BIN1 locally

¹Institut de Recherche en Infectiologie de Montpellier (IRIM), University of Montpellier, CNRS UMR 9004, Montpellier, France. ²CRBM, University of Montpellier, CNRS UMR 5237, Montpellier, France. ³Institut Curie, CNRS UMR 144, PSL Research University, Paris, France. ⁴Institut Curie, CNRS UMR 168, PSL Research University, Paris, France. ⁵Aix-Marseille Université, U1325 INSERM, DyNaMo, Turing center for living systems, Marseille, France. ⁶Dipartimento di Medicina Molecolare e Biotecnologie Mediche, Università degli Studi di Napoli Federico II, Naples, Italy. ⁷Université Paris-Saclay, CEA, CNRS, Institute for Integrative Biology of the Cell (I2BC), Gif-sur-Yvette, France. ⁸Grenoble Alpes University, INSERM U1216, Grenoble Institut Neurosciences, Grenoble, France. ⁹Department of Translational Medicine, IGBMC, U1258, UMR7104 Strasbourg University, Collège de France, Illkirch, France. ¹⁰These authors contributed equally: Cécile Gauthier-Rouvière, Stéphanie Miserey. ✉e-mail: laura.picas@irim.cnrs.fr; cecile.gauthier@crbm.cnrs.fr; Stephanie.Miserey@curie.fr

increases the PI(4,5)P$_2$ density to recruit dynamin on membranes selectively[28]. Moreover, the PI domain controls a conformational switch that facilitates BIN1 SH3 domain interaction with dynamin[23,29], and to other BIN1 partners such as the neuronal Wiskott-Aldrich syndrome (N-WASP) protein that regulates actin polymerization through the Arp2/3 complex[16,17,30]. Furthermore, in vitro and in cellulo approaches have shown that neuronal and muscle-specific BIN1 isoforms directly bind F-actin[17,31]. However, it is unknown how BIN1 remodels the plasma membrane/actin interface and whether it fulfils a functional role in forming actin-rich protrusion.

Here, we report that BIN1 is a multi-functional platform to promote filopodia formation in myoblasts. Our results show that BIN1 assembles actin bundles in vitro. We also identified additional BIN1-binding partners, including IRSp53 and ezrin, during filopodia formation. We show that BIN1 regulates the membrane-to-cortex mechanics, and it is required to recruit active phosphorylated ezrin at filopodia. Thus, we propose that BIN1 constitutes a functional membrane and actin-remodeling scaffold that, unexpectedly, as an N-BAR domain protein, orchestrates antagonistic membrane topologies at the cell cortex, such as the formation of filopodia.

## Results

### BIN1 expression is necessary and sufficient for filopodia formation

The association of different BIN1/Amphiphysin2 isoforms with F-actin and actin regulators such as the neuronal Wiskott-Aldrich syndrome (N-WASP)[16,17,31], prompt us to investigate if BIN1 proteins participate in actin-rich structure formation. Therefore, we expressed in HeLa cells GFP and several GFP-tagged amphiphysin isoforms: amphiphysin1 (typically associated with clathrin-coated structures)[32], the largest BIN1 isoform (BIN1 iso1, expressed in neurons), and the muscle-specific isoform BIN1 (BIN1 iso8), which contains the in-frame exon 11 encoding a polybasic motif binding phosphoinositides (PIs)[33], and stained F-actin. We observed that BIN1 iso8 drastically enhanced filopodia density, whereas BIN1 iso1 and Amphiphysin1 led to a low to null increase in filopodia density, respectively (Fig. 1a).

To inquiry into the molecular mechanisms of BIN1-mediated filopodia formation, we turned out to myoblast cells because actin protrusions have been implicated in the adhesion and fusion of muscle cells[20,34] and BIN1 expression increases during myoblast differentiation (Fig. 1b)[26]. To this end, we generated C2C12 myoblast expressing specific short interfering RNA (shRNA) against Bin1 or luciferase (CTRL shRNA) to ensure a homogenous knock-down of Bin1 in the myoblast cell population. In undifferentiated Bin1 shRNA C2C12 cells, depletion of BIN1 expression was almost complete (Fig. 1c). We then performed a detailed analysis of the cellular morphology of control and C2C12 Bin1 shRNA myoblasts by scanning electron microscopy (SEM) either under proliferative conditions and at 80–90% confluence to favor the formation of cell-cell junctions preceding myoblast differentiation and fusion. Furthermore, BIN1 depletion was associated with reduced membrane extensions between cells and a smoother dorsal plasma membrane (Fig. 1d-contacting cells and Supplementary Fig. 1a-isolated cells). This phenotype was evident at the cell periphery and cell-cell junctions, indicating that BIN1 might participate in adjoining myoblasts' intercellular zippering. Conversely, we observed a dense interface of filopodia interconnecting two or more adjacent myoblasts in control cells. In Bin1 knock-down cells, the filopodia densities are significantly decreased (3.6-fold reduction in filopodia density, Fig. 1e). These results show that in myoblasts, BIN1 depletion affects the morphology and the number of filopodia at cell-cell junctions and at the cell periphery (i.e., away from intercellular junctions).

Next, we analyzed BIN1 localization by multi-color life-cell imaging of C2C12 cells co-expressing Lifeact-mCherry and BIN iso8. We found that at the cell periphery BIN1 co-localized with actin at filopodia (Fig. 1f). Furthermore, kymograph analysis of the GFP-BIN1 signal showed that it often precedes that of Lifeact-mCherry (blue arrowheads, Fig. 1f). Moreover, BIN1 signal is present during the initiation, extension, and retraction of filopodial actin filaments (Fig. 1f).

Altogether, these data show that BIN1 expression promotes filopodia formation and that BIN1 knock-down decreases these F-actin rich structures.

### BIN1 promotes actin bundling

BIN1 iso1 was previously reported to bundle actin in vitro[31] and its N-BAR and SH3 domains share a strong homology with BIN1 iso8[33]. Thus, we sought to determine if the muscle-specific BIN1 iso8 also has the ability to associate and remodel F-actin. First, we performed a high-resolution analysis of BIN1 and F-actin using structured illumination microscopy (SIM) imaging in C2C12 cells. The endogenous BIN1 staining (Fig. 2a) or expression of GFP-BIN1 iso8 (Fig. 2b) revealed that BIN1 is localized at the base of filopodia and displays a discontinuous labeling along these F-actin-rich structures. Consequently, we determined whether BIN1 localization in filopodia is related to its association with actin. Thus, we performed co-immunoprecipitation experiments with GFP-tagged amphiphysin1, BIN1 iso1, and BIN1 iso8 (Fig. 2c). Immuno-blotting showed that in cellulo, only BIN1 iso8 appears associated with actin. Next, we determined the contribution of the BIN1 N-terminal N-BAR and the C-terminal SH3 domain for its association with actin in cellulo (Fig. 2d). To this end, we tested by co-immunoprecipitation the interaction with actin of the BIN1 D151N mutant, which carries a mutation in the N-BAR domain, which interferes with the curvature-sensing abilities and capacity to oligomerize[35], and of the BIN1 ΔSH3 mutant, which displays a truncated SH3 domain shown to prevent its interaction with cellular effectors[24] or N-WASP[16]. As shown in Fig. 2d, both mutants associate with actin in cellulo.

The co-localization and association of BIN1 iso8 with actin in cellulo prompted us to determine the potential ability of BIN1 iso8 to remodel F-actin in vitro using an assay based on purified actin and BIN1. We showed that BIN1 promotes actin filament-bundling (Fig. 2f). Remarkably, the number of actin bundles increased with increasing BIN1 concentrations, although we already observed a bundling effect at 0.1 μM of BIN1 with 1 μM of actin (Supplementary Fig. 1b), as compared to the actin-bundling activity reported for the BIN1 iso1 (≥0.25 μM according to ref. 31). Furthermore, we detected the similar actin-bundling activity by adding each of the BIN mutants separately (Fig. 2f). Altogether, these data show that mutations at the N-BAR and SH3 domain truncation do not perturb the F-actin association and bundling ability of BIN1.

Next, we analyzed the effect of the N-BAR and SH3 domain of BIN1 on filopodia formation in cellulo (Fig. 2g–k and Supplementary Fig. 1c). Therefore, GFP, GFP-BIN1 iso8, or GFP-tagged BIN1 D151N and BIN1 ΔSH3 mutants were co-expressed with Lifeact-mCherry to monitor the density and lifetime of actin-rich protrusions. As expected, expression of GFP-BIN1 iso8 in C2C12 cells increased filopodia density compared to control conditions (GFP alone). However, we did not detect a significant increase in their density with the D151N or ΔSH3 BIN1 mutants (Fig. 2g, h). We observed an increase in the % of static filopodia (i.e., filopodia that do not collapse for a period of time ≥120 s) upon BIN1 wild-type expression and the two BIN1 mutants (Fig. 2I), in agreement with their similar association to and bundling of F-actin (Fig. 2d–f). However, we did not detect differences in the mean lifetime (i.e., the time until filopodia collapse) of dynamic filopodia for any of the conditions tested (Fig. 2j). Furthermore, we observed that expression of the human form of BIN1 iso8 reestablished filopodia density on mouse shRNA Bin1 knock-down C2C12, whereas this was not the case of the D151N and ΔSH3 mutants (Fig. 2k).

These data indicate that the N-BAR and SH3 domains are likely to contribute to BIN1-induced filopodia formation, possibly favoring the association with BIN1-binding partners.

### IRSp53 is required for BIN1-mediated filopodia formation

To evaluate the contribution of potential BIN1 partners in filopodia formation, we performed a proteomic analysis using GFP-BIN1 iso8 to identify its binding partners (Supplementary Fig. 2). We found already known BIN1 partners (in gray Supplementary Fig. 2): dynamin 1 and 2[24,26] and actin (this study and ref. 17), and unknown partners (in yellow Supplementary Fig. 2)

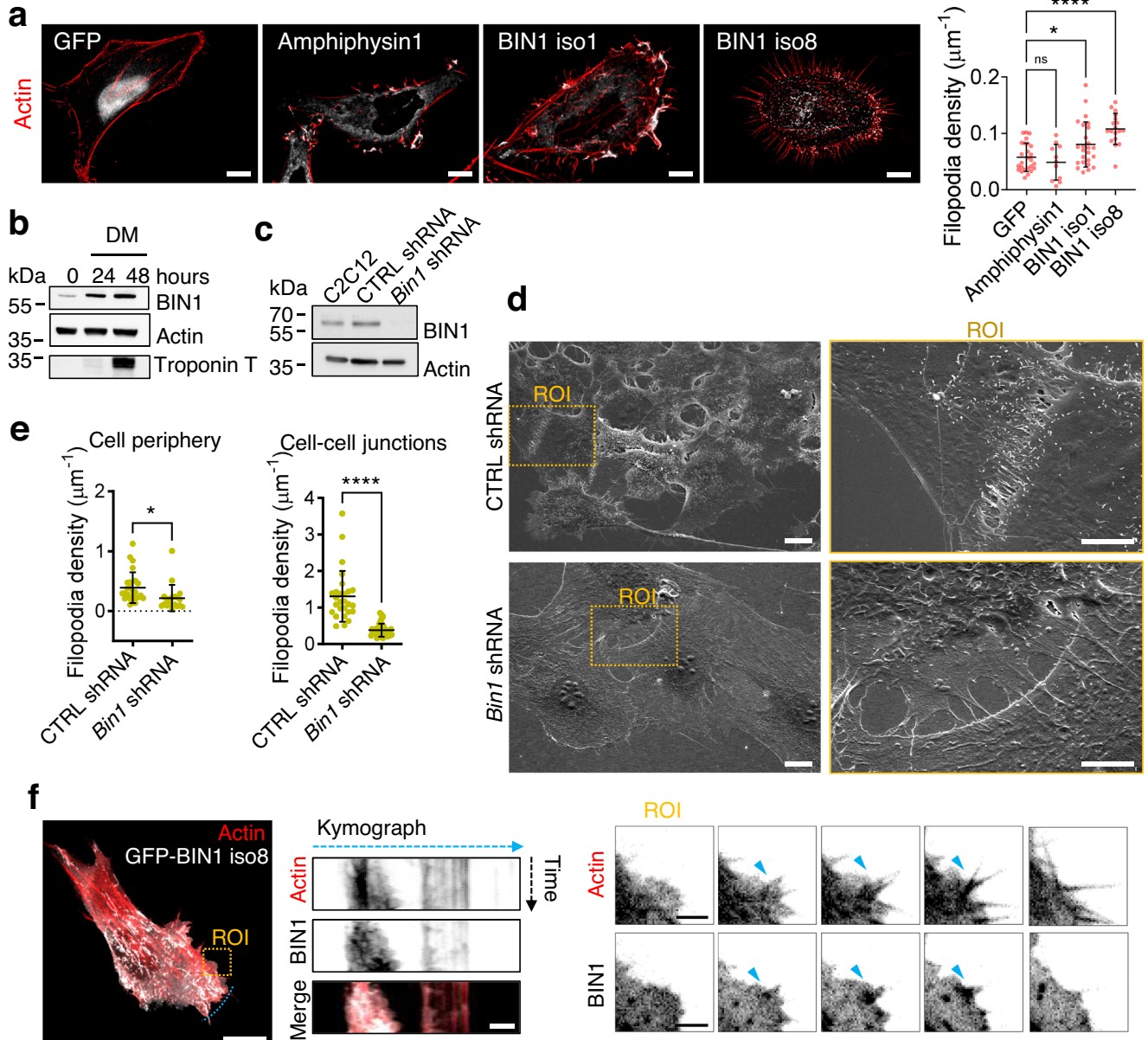

**Fig. 1 | BIN1 expression promotes the formation of filopodia. a** Wild-field deconvoluted images of HeLa cells transfected with GFP or GFP-tagged Amphiphysin1, BIN1 iso1 and BIN1 iso8 (in gray) and stained for F-actin (phalloidin, red). Scale bar, 10 μm. Quantification of the density of HeLa cells expressing GFP or GFP-tagged Amphiphysin1, BIN1 iso1 and BIN1 iso8. Number of cells: *n* = 26, 11, 25 and 18, respectively from three independent experiments. **b** Western-blot analysis of the endogenous expression of actin, BIN1, and Troponin T in C2C12 myoblasts in growth medium (0 h, undifferentiated) and grown in differentiation medium (DM) at 24 h and 48 h. **c** Western-blot analysis of the endogenous expression of actin and BIN1 on parental, CTRL shRNA (i.e., Luciferase) and *Bin1* shRNA C2C12 myoblast cells. **d** Scanning electron microscopy images of CTRL shRNA and *Bin1* shRNA C2C12 myoblasts cultured in growth factor-containing medium. Scale bar, 10 μm

and 5 μm for the magnified images. **e** Quantification of the number of filopodia per cell at the cell periphery and at intercellular junctions measured from the SEM images. Cells: *n* = 25, 19 and 27, 38 for CTRL and *Bin1* shRNA cells at the cell periphery and cell-cell junctions, respectively. **f** Snap-shots of spinning disk life cell imaging (500 ms exposure, during 60 s. Total time acquisition = 120 s) of C2C12 myoblasts co-transfected with Lifeact-mCherry (red) and GFP-BIN1 iso8 (gray) at *t* = 0 s. Kymograph analysis along the blue dashed line in the corresponding image at t = 0 s, highlight the recruitment and binding of BIN1 iso8 to filopodia. Scale bar, 10 μm. Scale bar in kymograph and ROI, 2 μm. Bottom, representative time-lapse snapshots from the ROI region showing the localization of BIN1 iso8 on F-actin during filopodia formation in C2C12 cells, as highlighted by the blue arrowheads. Error bars represent s.d.; ANOVA test: n.s > 0.1, *P < 0.05, ****P < 0.0001.

such as IRSp53 and ezrin. Interestingly, IRSp53 is involved in the Cdc42-dependent formation of filopodia[9,36,37]. Expression of GFP-BIN1 iso8 and IRSp53-mCherry on C2C12 myoblasts showed that both proteins localize together on >75% of IRSp53-positive filopodia (Fig. 3a, b). However, this trend was decreased in the presence of the BIN1 D151N or ΔSH3 mutants. Furthermore, we confirmed by immunofluorescence labeling that endogenous IRSp53 and of GFP-BIN iso8 co-localize at filopodia (Supplementary Fig. 3). Co-immunoprecipitation experiments using BIN1 iso8 and the

BIN1 D151N or ΔSH3 mutant showed that only wild-type BIN1 and the D151N mutant associate with IRSp53 in cellulo (Fig. 3c), suggesting that the SH3 domain of BIN1 is likely to mediate the interaction with IRSp53.

Next, we determined the role of IRSp53 in BIN1-mediated filopodia formation in C2C12 cells (Fig. 3d–f). As expected, IRSp53 knock-down in C2C12 cells led to a decrease in filopodia density. While the expression of GFP-BIN1 iso8 in CTRL siRNA C2C12 promoted an increase in filopodia formation, this was not the case under IRSp53 knock-down, therefore,

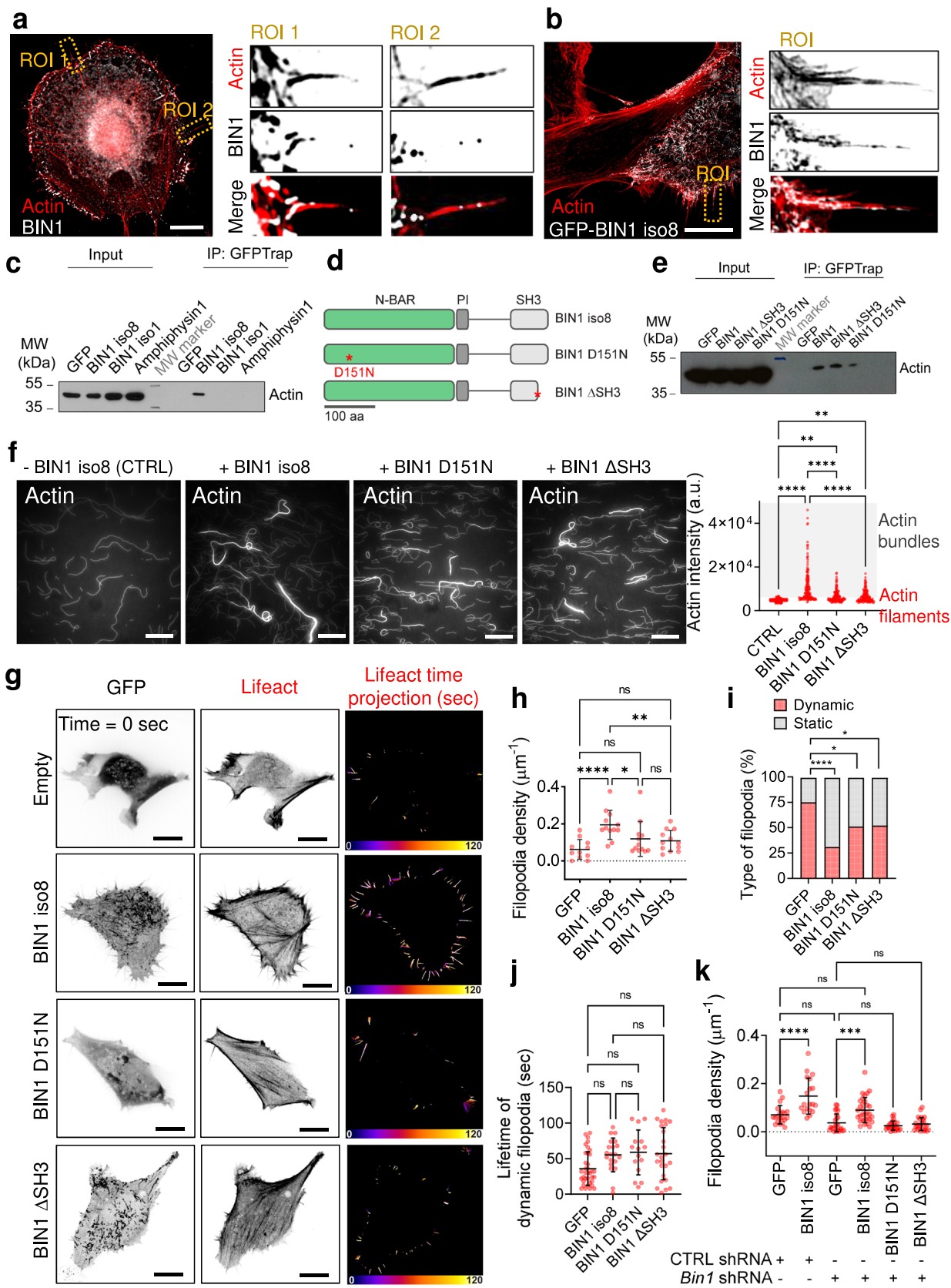

suggesting that BIN1-mediated filopodia formation requires the I-BAR protein IRSp53.

The topology at the neck of filopodia is compatible with the binding of proteins with positive curvature BAR-domain sensors like BIN1[12].

Therefore, we inquire whether BIN1 iso8 can be recruited to negatively-curved membranes induced by IRSp53 using an in vitro reconstituted assay and confocal microscopy (Fig. 3e), as previously reported[38]. We generated 5% mole PI(4,5)P$_2$-containing giant unilamellar vesicles (GUVs) doped

**Fig. 2 | BIN1 interacts with F-actin in vitro and in cellulo. a** C2C12 myoblasts stained for endogenous BIN1 (C99D antibody, gray) and F-actin (phalloidin, red). Magnified images of two representative regions of interest (ROIs). Scale bar, 10 μm. **b** SIM images of C2C12 myoblasts expressing GFP-BIN1 (gray) stained for F-actin (phalloidin, red) and magnified images of two representative ROI in the corresponding image. Scale bar, 10 μm. **c** GFP pull-downs using extracts from HeLa cells expressing either GFP, or GFP-BIN1 iso8, GFP-BIN1 iso1 and GFP-Amphiphysin1. Actin was detected by western blotting. HeLa cells were used to provide an unbiased cellular context. IP = immunoprecipitate. **d** Domain representation (N-BAR, phosphoinositide-binding motif, PI, and SH3 domains) of BIN1 iso8 full-length. Stars highlight the D151N mutant at the N-BAR and the stop codon of the BIN1 ΔSH3 mutant. **e** GFP-Trap pull-downs using extracts from C2C12 myoblasts expressing either GFP alone or fused with BIN1 iso8, BIN1 ΔSH3 or BIN1 D151N. Actin was revealed by western blotting. IP = immunoprecipitate. **f** In vitro assay with purified actin and BIN1 iso8 and BIN1 D151N and BIN1 ΔSH3 mutants. TIRF images showing pre-polymerized F-actin filaments in the absence (-BIN1 iso8, CTRL) or in the presence of 1 μM of BIN1 iso8 and BIN1 mutants and the corresponding quantification of the BIN1 bundling activity as denoted by the actin intensity in each condition. Scale bar, 20 μm. **g** Snap-shots of spinning disk life cell imaging at t = 0 of C2C12 cells co-transfected with mCherry-Lifeact and either GFP (empty), GFP-BIN1 iso8, GFP-BIN1 D151N or GFP-BIN1 ΔSH3 (inverted LUT images of the actin and GFP signal). Time projection of the filopodia tracks (500 ms exposure, during 60 s. Total time acquisition = 120 s) obtained from the corresponding Lifeact images. Scale bar, 10 μm. Fire LUT color scale is 120 s. **h** Quantification of filopodia density; cells: n = 11, 12, 12 and 12 for GFP, BIN1 iso8, D151N and ΔSH3 mutants, respectively. **i** % of dynamic (filopodia that collapsed for a period ≤ 120 s, red) and static (filopodia that do not collapse for a period ≥ 120 s, gray) filopodia; Number of filopodia: n = 49, 64, 29, and 48 for GFP, BIN1 iso8, D151N and ΔSH3 mutants, respectively. Chi-square test: *P < 0.1, ****P < 0.0001. **j** Lifetime of dynamic filopodia (in seconds); Number of filopodia: n = 36, 20, 15, and 25 for GFP, BIN1 iso8, D151N and ΔSH3 mutants, respectively. **k** Quantification of filopodia density; Cells: n = 20, 19, 25, 31, 25, and 25 for CTRL shRNA and Bin1 shRNA C2C12 cells co-transfected with either GFP, GFP-BIN1 iso8, GFP-BIN1 D151N or GFP-BIN1 ΔSH3 and Lifeact-mCherry, respectively. Error bars represent s.d.; ANOVA test: n.s > 0.1, *P < 0.05, **P < 0.01, ***P < 0.001, ****P < 0.0001.

with Oregon Green 488 DHPE, as PI(4,5)P$_2$ is a central phosphoinositide for BIN1 and IRSp53 membrane interactions[28,39]. As expected, the addition of the full-length IRSp53 led to the formation of inward membrane tubes on PI(4,5)P$_2$-GUVs. The addition of recombinant alexa fluor A647-labeled BIN1 iso8 showed that it accumulates to membrane indentations that are generated by IRSp53 (Fig. 3g). To confirm that the presence of IRSp53 mediates BIN1 localization to the base of inward membrane tubes, we generated dark PI(4,5)P$_2$-containing GUVs and monitored the co-localization of recombinant alexa fluor A488-labeled IRSp53 and alexa fluor A647-labeled BIN1 iso8 (Fig. 3h). Our results confirmed that IRSp53 allows BIN1 recruitment at the base of negatively-curved membrane structures.

## BIN1-mediated recruitment of ezrin participates in filopodia formation

Interestingly, our proteomic study identified additional BIN1 interactors, such as the members of the ezrin-radixin-moesin (ERM) protein family (Supplementary Fig. 2), previously found to accumulate at the shaft of filopodia and to co-localize with IRSp53 on microvilli[7,40]. To study if specificity exists between different BIN isoforms and the two ERM members ezrin and moesin, we performed co-immunoprecipitation experiments using several amphiphysin isoforms: amphiphysin1, BIN1 iso1 and BIN1 iso8. Our results showed that only BIN1 iso8 associated with ezrin in cellulo (Fig. 4a), while both BIN1 iso1 and iso8 isoforms associate with moesin (Supplementary Fig. 4a). We also found that only the BIN1 D151N mutant, but not the ΔSH3 mutant, associates with ezrin (Fig. 4b), suggesting that the SH3-containing C-terminal domain of BIN1 mediates the interaction with ezrin.

Next, we tested the association of BIN1 and ezrin during filopodia formation. We performed multi-color time-lapse live-cell imaging of C2C12 cells co-expressing mCherry-ezrin together with either GFP-BIN1 iso8 and the ΔSH3 mutant that, according to our biochemical assay does not associate with ezrin (Fig. 4c and Supplementary Fig. 4b). Spinning-disk images showed that BIN1 and ezrin co-localize at the cell periphery (Fig. 4c, d). Detailed analysis of the dynamics of ezrin and BIN1 also showed that both proteins co-localized during filopodia formation (Fig. 4d, magnified ROI in yellow and kymograph analysis along the blue-dashed line in C, and Fig. 4e). In addition, co-expression of ezrin with BIN1 potentiated the formation of filopodia compared to BIN1 alone (Fig. 4e), suggesting that BIN1 and ezrin might cooperate in this process. Conversely, an absence of co-localization and of effect on filopodia density was observed between ezrin and the BIN1 ΔSH3 mutant, indicating that BIN1-ezrin mediated filopodia formation might require ezrin association to BIN1 via the SH3 domain of BIN1 (Supplementary Fig. 4b).

Ezrin displays different conformational states, including a cytosolic closed-conformation, which results from the intramolecular interaction of the N-terminal FERM domain with the C-terminal ERM-associated domain (C-ERMAD), a membrane-bound opened-conformation, which requires a sequential activation through the interaction with PI(4,5)P$_2$, a phosphorylated conformation, which displays a phosphorylation of a conserved threonine in the actin-binding site of the C-ERMAD (T567, in ezrin) and finally, the interaction of the C-ERMAD with F-actin[6]. Interestingly, we previously showed that BIN1 clusters PI(4,5)P$_2$ to recruit PI(4,5)P$_2$-interacting downstream partners on membranes[28]. Therefore, we analyzed if PI(4,5)P$_2$, a key phosphoinositide for the recruitment of actin regulatory proteins promoting actin polymerization and filopodia formation at the plasma membrane[41,42] could participate in ezrin membrane recruitment by BIN1. We confirmed that filopodia positive for BIN1 are highly enriched in PI(4,5)P$_2$ (Supplementary Fig. 4c). Next, using an in vitro reconstituted assay consisting of supported lipid bilayers doped with 5% PI(4,5)P$_2$, we quantified the binding of recombinant wild-type ezrin and its phospho-mimetic form (ezrin-T567D) in the presence of recombinant BIN1 iso8 (Fig. 4f). As a control we used amphiphysin 1, a protein that does not interact with ezrin, but that mediates PI(4,5)P$_2$ clustering[28]. We estimated the relative binding of ezrin, wild type or T567D, from the ratio between the intensity of ezrin proteins bound on membranes in the presence of BIN1 iso8 or of amphiphysin1, normalized by the intensity of ezrin proteins in the absence of BIN1 or amphiphysin1. We obtained a ~2.5-fold and 1.5-fold increase in the relative membrane binding of ezrin and ezrin-T567D, respectively, in the presence of BIN1 and PI(4,5)P$_2$ (Fig. 4f). In agreement with the results in Fig. 4a, amphiphysin1 did not affect ezrin binding to membranes. Altogether these observations suggest that BIN1-mediated ezrin recruitment on membranes is probably mediated by protein-protein interactions (Fig. 4b) on PI(4,5)P$_2$ interfaces.

To further understand the molecular mechanisms of BIN1/ezrin interaction at the cell cortex, we investigated the impact of Bin1 knock-down on ezrin and phosphorylated ezrin (phospho-ezrin) expression (Fig. 4h) and localization (Fig. 4i). We observed a two-fold increase of phosphorylated ezrin in Bin1 shRNA C2C12 cells, but no modification of the total pool of ezrin (Fig. 4h). In addition, we observed that phospho-ezrin appears enriched at filopodia in control C2C12 cells (Fig. 4i). However, such enrichment was not observed in C2C12 Bin1 shRNA myoblasts. These results indicate that although the total level of phospho-ezrin is increased, the cellular organization of phospho-ezrin appears affected in Bin1 knock-down myoblasts. This suggests that BIN1 might regulate the localization and activation of ezrin at the plasma membrane.

## BIN1 regulates the cell cortex architecture of myoblast

In myoblast cells, BIN1 expression increases during differentiation (Fig. 5a)[24,26], and its inhibition is associated with impaired myotube formation[25,26], data that we also confirmed (Supplementary Fig. 5). Furthermore, we observed that the expression of different differentiation

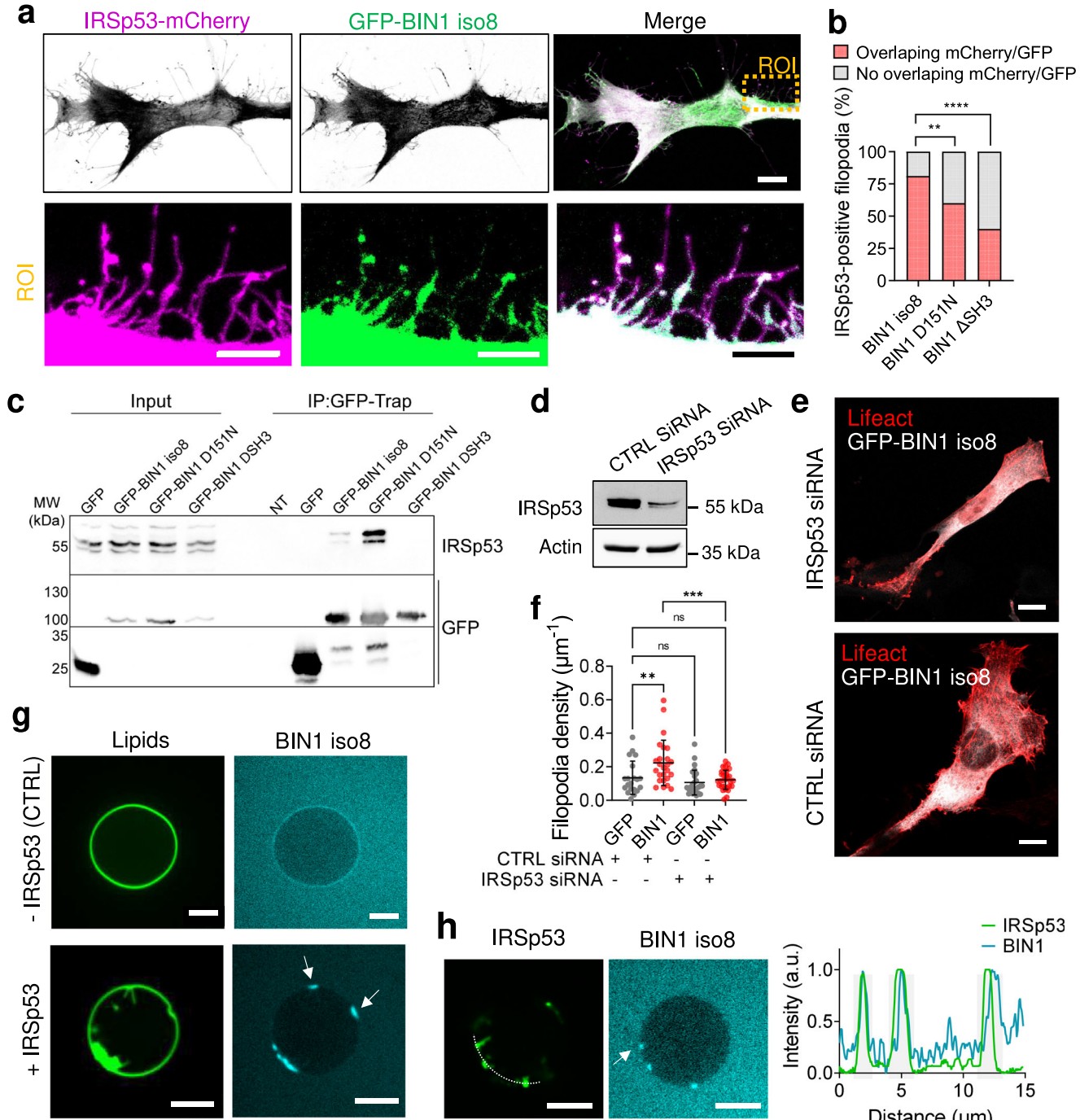

**Fig. 3 | BIN1 associates with IRSp53 on membrane protrusions. a** Confocal images of C2C12 cells transfected with GFP-BIN1 iso8 (in green) and IRSp53-mCherry (magenta). Magnified image of the corresponding ROI. Scale bar, 10 µm. Scale bar ROI, 5 µm. **b** % of IRSp53-positive filopodia displaying an overlapping of mCherry/GFP signal (red) or no overlapping of signals (gray) upon co-expression of GFP-BIN1 iso8, GFP-BIN1 D151N or GFP-BIN1 ΔSH3 and IRSp53-mCherry. Number of filopodia: $n$ = 377, 360 and 475, respectively. Chi-square test: n.s > 0.1, ****$P$ < 0.0001. **c** GFP-Trap pull-downs from extracts of HeLa cells transfected with plasmids encoding GFP, GFP-BIN1 iso8 and the BIN1 mutants D151N and ΔSH3. IRSp53 was revealed by western-blotting ($n$ = 3). **d** Western-blot analysis of the endogenous expression of actin and IRSp53 on CTRL siRNA (i.e., Luciferase) and IRSp53 siRNA C2C12 cells. **e** Z-projected confocal images showing the actin

organization of CTRL siRNA and IRSp53 siRNA C2C12 cells co-transfected with GFP-BIN1 iso8 (gray) and Lifeact-mCherry (red). **f** Quantification of filopodia density; $n$ = 22, 26, 26 and 27 for CTRL siRNA and IRSp53 siRNA C2C12 cells co-transfected with either GFP or GFP-BIN1 and Lifeact-mCherry respectively. Error bars represent s.d.; ANOVA test: n.s > 0.1, **$P$ < 0.01, ***$P$ < 0.001. **g** Representative confocal images of 0.1 µM BIN1 binding (cyan) to control (only BIN1) or in the presence of 0.3 µM IRSp53-induced tubules on PI(4,5)P$_2$-containing GUVs (green). Scale bar, 5 µm. **h** Representative confocal images of 0.1 µM BIN1 binding (cyan) to PI(4,5)P$_2$-containing GUVs (dark) in the presence of 0.3 µM IRSp53 (green). Profile analysis of BIN1 and IRSp53 signal along the dashed line in the corresponding image. Scale bar, 5 µm.

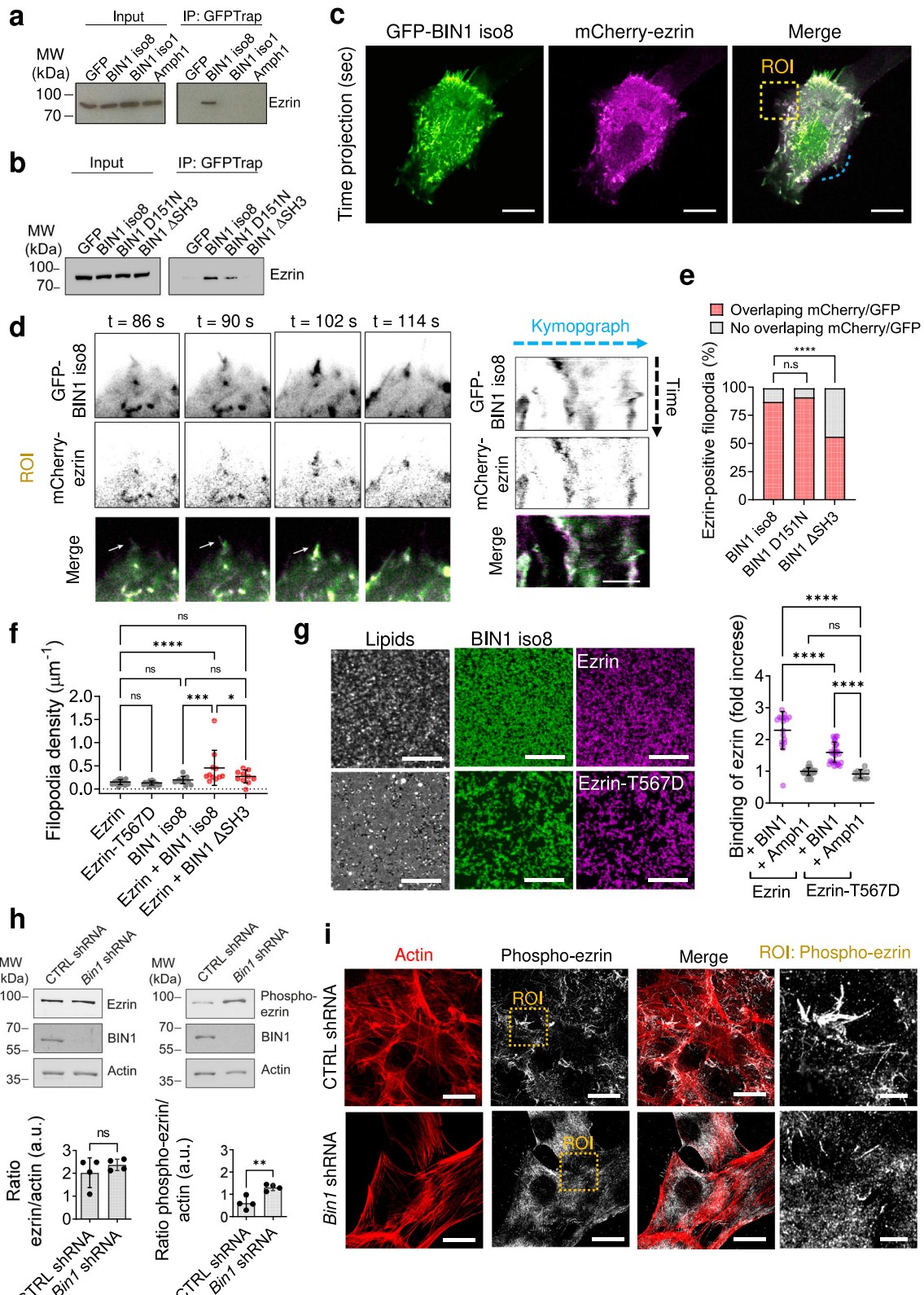

markers was not affected in *Bin1* shRNA C2C12 (Supplementary Fig. 5), suggesting a potential role of BIN1 during the fusion stages of myoblast cells.

Previous works showed that the SH3 domain of BIN1 mediates the interaction with its downstream partner dynamin[23,43], a GTPase well known to bundle actin filaments and having an important role during filopodia

formation promoting myoblast fusion in vivo[19,20,44,45]. Therefore, we confirmed by immunofluorescence that endogenous BIN1 co-localizes with GFP-dynamin2 at actin-rich structures on C2C12 cells (Supplementary Fig. 6). In contrast, the density and morphology of dynamin2-positive actin-rich protrusions appeared affected in C2C12 *Bin1* shRNA myoblasts (Supplementary Fig. 6). Collectively, supporting that, in addition to their

**Fig. 4 | BIN1 regulates ezrin localization on membranes. a** GFP-Trap pull-downs from extracts of HeLa cells transfected with plasmids encoding GFP, GFP-tagged amphiphysin1, BIN1 iso1 and BIN1 iso8. Ezrin was revealed by western-blotting. **b** GFP-Trap pull-downs using extracts from C2C12 myoblasts expressing either GFP alone or fused with BIN1 iso8, BIN1 ΔSH3 or BIN1 D151N. Ezrin was revealed by western-blotting. **c** Time projected spinning disk movies (500 ms exposure, during 60 s. Total time acquisition = 120 s) of C2C12 cells co-expressing mCherry-ezrin (magenta) and GFP-BIN1 iso8 (green). **d** Time-lapse snapshots from the corresponding yellow ROI showing the co-localization of BIN1 and ezrin during filopodia formation (white arrows). Kymograph analysis along the blue dashed line. Scale bar, 10 μm. Scale bar in kymographs, 1 μm. **e** % of ezrin-positive filopodia displaying an overlapping of mCherry/GFP signal (red) or no overlapping of signals (gray) upon co-expression of GFP-BIN1 iso8, GFP-BIN1 D151N or GFP-BIN1 ΔSH3 and mCherry-ezrin. Number of filopodia: $n$ = 384, 191 and 146, respectively. Error bars represent s.d.; Chi-square test: n.s > 0.1, ****$P$ < 0.0001. **f** Quantification of filopodia density; $n$ = 19, 17, 12, 10, and 10 for ezrin, ezrin-T567D, BIN1, ezrin + BIN1 iso8, and ezrin + BIN1 ΔSH3 mutant, respectively. $n$ = number of cells from

live cell imaging experiments. Error bars represent s.d.; ANOVA test: n.s > 0.1, *$P$ < 0.05, ***$P$ < 0.001, ****$P$ < 0.0001. **g** Confocal images of the co-localization of recombinant BIN1-Alexa647 (magenta), ezrin or phospho-mimetic ezrin (T567D) tagged with Alexa488 (green) and TopFluor-TMR-PI(4,5)P$_2$ (cyan) on supported lipid bilayers containing 5% of PI(4,5)P$_2$. Scale bar, 3 μm. Fold increase in the binding of recombinant ezrin and ezrin-T567D on supported lipid bilayers containing 5% PI(4,5)P$_2$ in the presence of BIN1 or amphiphysin1 (Amph1). **h** Western-blot analysis of the endogenous expression of ezrin and phosphorylated ezrin (phospho-ezrin), BIN1 and actin on cell extracts from CTRL (blue) and *Bin1* shRNA (yellow) C2C12 myoblasts, and the corresponding quantification of the signal of each protein normalized by actin. **i** CTRL and *Bin1* shRNA C2C12 myoblasts cultured in growth factor-containing medium and stained for endogenous phosphorylated ezrin (phospho-ezrin, yellow) and F-actin (phalloidin, cyan). Inset, high-magnification images showing the localization of endogenous phosphorylated ezrin. Maximum intensity projected airyscan images. Scale bar, 10 μm and 5 μm, respectively.

---

shared role in membrane remodeling, BIN1 and its partner dynamin2 can also interact on actin-rich protrusions.

Furthermore, ezrin, one of our BIN1-interacting partners during filopodia formation, was recently shown to be involved in myoblast differentiation and fusion[46]. Therefore, to investigate if the association of BIN1/ezrin on filopodia is maintained during myoblast differentiation, we analyzed the expression (Fig. 5a, b) and co-localization of endogenous BIN1 and the active phosphorylated form of ezrin (phospho-ezrin) in proliferating C2C12 cells (Day 0), i.e., ≤80–90% cell confluency in growth medium, and at 1 day of differentiation (Day 1), i.e., 24 h in differentiation medium. We observed that during differentiation BIN1 expression is conveyed with an increase in phospho-ezrin expression (Fig. 5a, b). Furthermore, BIN1 appears colocalized with phospho-ezrin on filopodia both under proliferating and differentiation conditions (Fig. 5c). Thus, pointing out that the expression and membrane localization of BIN1 and the active form of ezrin must be finely regulated during myoblasts differentiation.

As a central protein linking the actin cytoskeleton to the inner leaflet of the plasma membrane, ezrin is an important regulator of the mechanical cohesion of the membrane/actin interface[8,47,48]. We thus assessed the cortex-to-membrane mechanics from the force required to form membrane tethers on myoblasts, as previously described[47]. To this end, we used atomic force microscopy (AFM) cantilevers coated with poly-L-lysine to pull membrane tethers from the plasma membrane of control and *Bin1* knock-down C2C12 cells (Fig. 5d). We measured the static tether force, $f_0$, which is required to hold a membrane tether at a constant height[49] (as detailed in the methods section). We found that the apparent static tether force is decreased in *Bin1* shRNA cells as compared to control conditions (Fig. 5d). In agreement, we also observed, using time-lapse phase-contrast videomicroscopy, the presence of prolonged retraction fibers similar to membrane tethers behind migrating *Bin1* knock-down cells (Fig. 5e: see yellow arrows pointing to membrane tethers). Therefore, suggesting that BIN1 regulates, either directly or through its association with ezrin, the stability of the plasma membrane/actin interface.

## Discussion

We unravel here an unexpected role of BIN1 in promoting filopodia formation in skeletal muscle cells, structures that play a crucial role in myoblast adhesion and fusion in mammalian cells and *Drosophila*[20,34]. The sequence of events that can be envisioned for the BIN1-mediated formation of filopodia at the myoblast cortex is presented in Fig. 6.

Whereas BIN1 was shown to promote positive membrane curvature[26,28], we show here that BIN1 could also regulate the formation of membrane protrusions that have a negatively curved membrane geometry. Indeed, BIN1 expression in skeletal muscle cells promotes filopodia formation, and, reciprocally, its knock-down decreases their density at the cell surface, including at intercellular cell-cell contacts. High-resolution microscopy revealed that BIN1 is present at the base and all along the

filopodia. Such an unexpected function was also reported in the case of PACSIN2/Syndapin II, an F-BAR domain protein shown to be involved in cellular protrusion formation[50], presumably through the binding to the positively curved membrane at the base and the tip of filopodia[51].

Our proteomic analysis revealed that BIN1 is associated with proteins having important functions during filopodia formation such as actin, dynamin, IRSp53, and the ERM proteins. We demonstrated that the generation of these filopodia by BIN1 requires IRSp53, an I-BAR protein that initiates the formation of filopodia[9,10,37] and is implicated in the generation and maintenance of filopodia during the fusion process in *Drosophila*[34]. IRSp53 acts as a membrane-curvature sensing platform for assembling protein complexes[52]. Our in vitro data with purified IRSp53, which is active in this condition in the absence of Cdc42[53], show that IRSp53 allows BIN1 recruitment at the base of negatively-curved membranes and that the SH3 domain of BIN1 is likely to participate in this process. BAR domain proteins are well known to generate stable lipid membrane microdomains[28,54]. BIN1 has a PI domain that targets it to membrane domains, and we previously showed that BIN1 induces the clustering of PI(4,5)P$_2$. IRSp53 directly binds PI(4,5)P$_2$-rich membranes and deforms them[39], and might also promote the formation of PI(4,5)P$_2$ clusters[54]. We thus could envision a positive feedback loop with both proteins favoring their respective recruitment at specific membrane sites, where IRSp53 recruitment, in turn, assists BIN1 localization at the base of filopodial structures. Moreover, these PI(4,5)P$_2$ clusters might facilitate the recruitment of proteins having PI(4,5)P$_2$ binding motifs, such as actin regulators involved in protrusion formation[41,42]. The recruitment of actin regulators could also be directly mediated by BIN1 and IRSp53-dependent protein-protein interactions. Among the BIN1 partners known to regulate actin remodeling are WASP and dynamin[19,20,30]. Moreover, the SH3 domain of IRSp53 allows it to interact with VASP[9], N-WASP[37], WAVE and mDia[55] or also EPS8, an actin-filament bundling and capping protein[56,57].

Actin filament-bundling and membrane interactions are both essential features for the shape maintenance of filopodium[4,5]. BIN1 could participate in F-actin bundling at least *via* two mechanisms. We discovered through in vitro assays a direct role of BIN1 in F-actin bundling, which combined with its direct association with actin, explains why BIN1 is closely connected with F-actin along filopodia. Moreover, the interaction of BIN1 with its downstream partner dynamin[23,43] is likely to enhance the formation of actin-rich protrusions via its multifilament actin-bundling ability required for efficient myoblast fusion[20]. Our study shows that other BIN1/amphiphysin family members can promote filopodia formation, although to a lesser extent. In vitro BIN1 iso8 already displays an actin-bundling ability at lower concentrations than the neuronal isoform (BIN1 iso 1[31]), which might explain the differences between the two isoforms in promoting filopodia formation in cellulo. However, other factors could also contribute to differences in the filopodia density phenotype observed. For instance, the selective targeting of BIN1 iso8 to the plasma membrane through its PI(4,5)P$_2$-binding motif[26], which is not present in BIN1 iso1, might enhance its

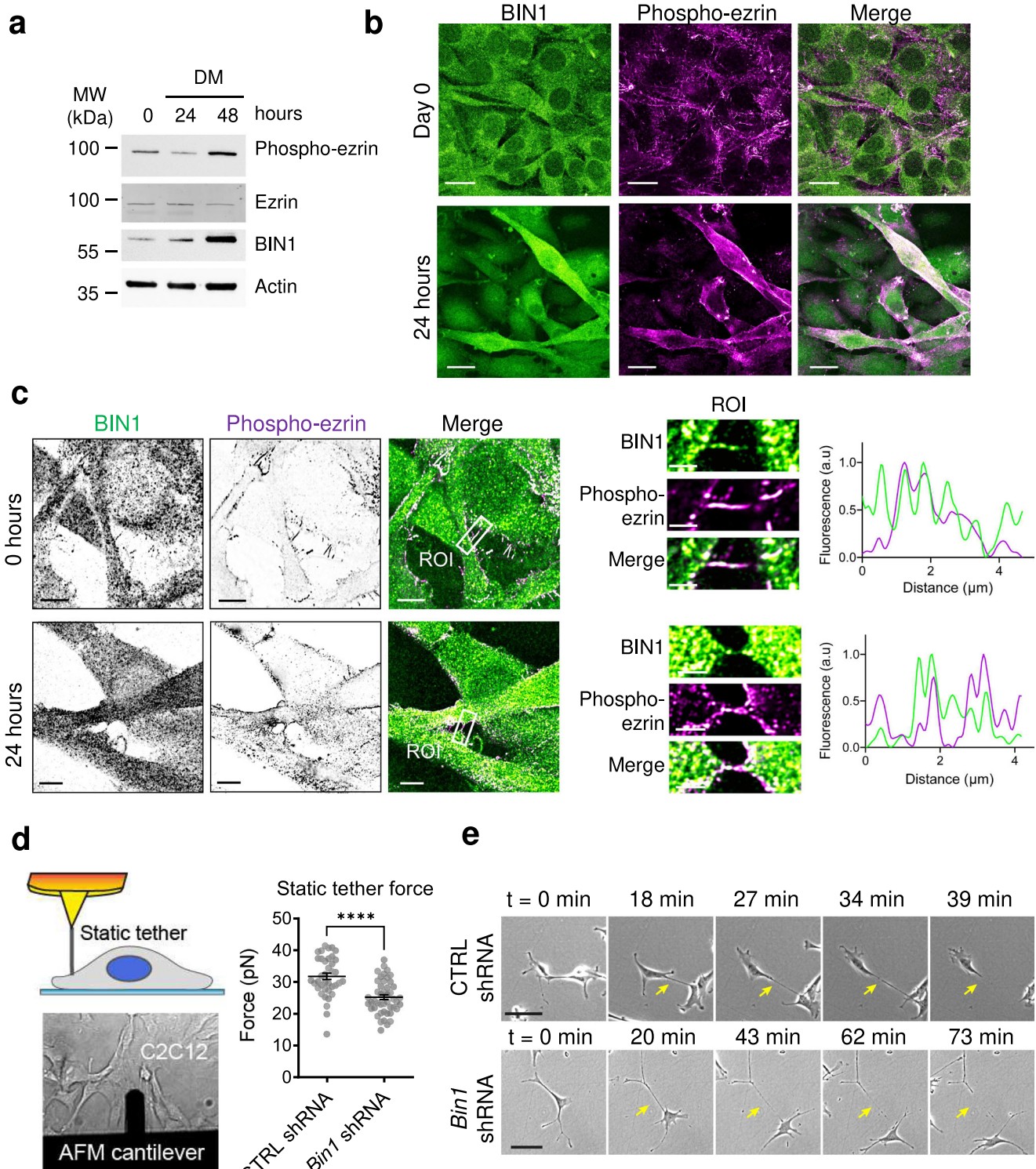

**Fig. 5 | BIN1 regulates the cell cortex architecture. a** Western-blot analysis of the endogenous expression of BIN1, ezrin and phosphorylated ezrin (phospho-ezrin) in C2C12 myoblasts under proliferative conditions (Day 0) or at 24 h and 48 h in differentiation medium (Day 1 and Day 2, respectively). **b** Maximum intensity projected confocal images of proliferative myoblasts (Day 0) or at 24 h in differentiation medium (Day 1) stained for endogenous BIN1 (C99D antibody, green) and phosphorylated ezrin (phospho-ezrin, magenta). Scale bar, 20 μm. **c** Airyscan images of C2C12 myoblasts at day 0 and day 1 stained for endogenous BIN1 (green) and phospho-ezrin (magenta). Cross-section along a representative filopodia (BIN1 in green, phospho-ezrin in magenta) highlighted by the white box and magnified in the ROI image. Scale bar is 5 μm, and 2 μm for the inset. **d** Schematic representation of a plasma membrane tether pulling assay using an AFM cantilever coated with poly-L-lysine on adherent C2C12 cells. Representative bright field image of the corresponding experimental setup. Static tether force (pN) was obtained with CTRL (blue) and *Bin1* shRNA (yellow) C2C12 myoblasts. *t*-test: *P* < 0.0001. **e** Snapshots of bright field images of CTRL and *Bin1* shRNA C2C12 myoblasts cultured in growth factor-containing medium at different time points. Yellow arrow highlights the retraction of retraction fibers formed during cell migration. Scale bar, 50 μm.

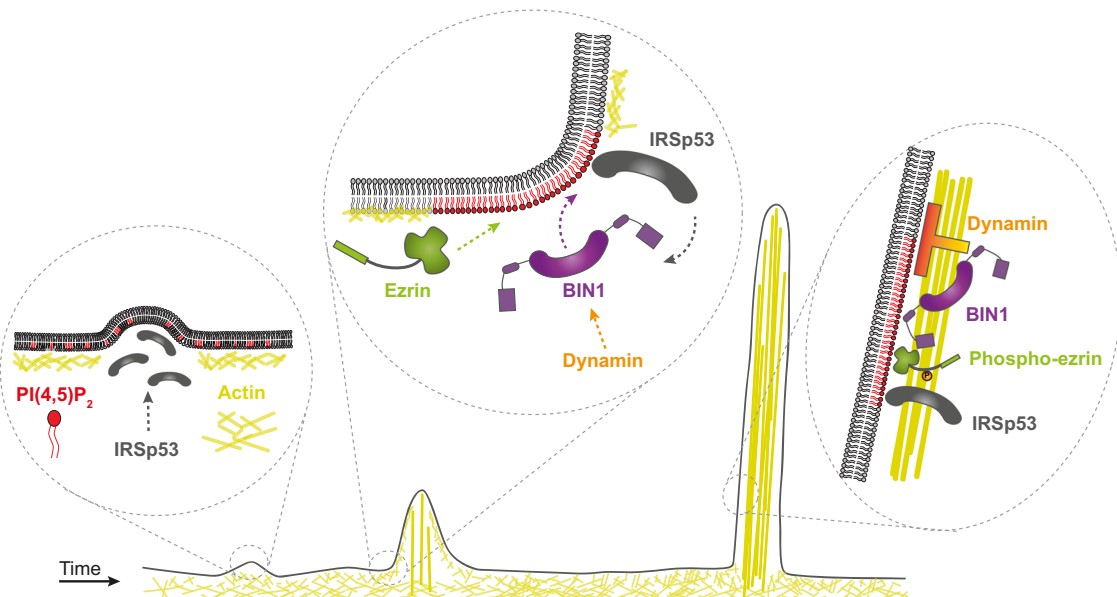

**Fig. 6 | Model of BIN1-mediated filopodia formation at the myoblast membrane.** The following steps are shown: 1) formation of BIN1-mediated filopodia requires an IRSp53-based actin complex leading to the initial evagination of the plasma membrane[10]. 2) Binding of BIN1 at the plasma membrane might led to the formation of a $PI(4,5)P_2$-rich interface[28]. This $PI(4,5)P_2$ enrichment might, in turn, facilitate the accumulation of $PI(4,5)P_2$-interacting proteins involved in the formation of actin-rich protrusions such as IRSp53, ezrin, and dynamin at the cell cortex. 3) The actin bundling ability of BIN1 and its association with phospho-ezrin, IRSp53 and dynamin[20] facilitates its localization during filopodia formation. The different molecular players involved in this process are: IRSp53 in dark gray, BIN1 in magenta, dynamin in orange, ezrin and phospho-ezrin in green, actin in yellow, and $PI(4,5)P_2$ in red.

function on $PI(4,5)P_2$-mediated processes such as actin remodeling[42]. Also, the contribution of other cellular factors, such as IRSp53, could not be excluded, and it would be interesting to address further its contribution to recruiting different BIN1/amphiphysin isoforms at negatively-curved membranes based on the homology of their SH3 domains[33].

Furthermore, our proteomic analysis of the BIN1 partners confirmed its association with dynamin and actin and, importantly, identified the ERM family proteins. BIN1 is the only amphiphysin member tested to associate with ezrin. We focused on the BIN1/ezrin interaction and function because ezrin is a membrane-actin linker that anchors F-actin to the plasma membrane at different types of membranes protrusions[58], and appears enriched in filopodia[59]. We showed that BIN1 and the phosphorylated active form of ezrin are co-localized in filopodia, and in the absence of BIN1, phosphorylated ezrin is no more enriched in filopodia. BIN1 could participate in ezrin recruitment through several non-exclusive mechanisms. First, BIN1 SH3 domain is crucial for its association with ezrin, suggesting direct protein-protein interaction. Secondly, ezrin binding could also be mediated by the IRSp53 protein which was shown to enrich ezrin on negatively curved membranes[8]. Local depletion of ezrin from the plasma membrane is required to initiate actin-driven protrusion formation[60]. Our study shows that BIN1-mediated filopodia formation is likely to occur once the protrusion is initiated, as it requires IRSp53 for its localization at negatively-curved membrane indentations. Thus, BIN1/ezrin association might enhance the membrane/actin interaction at later stages of filopodia formation. This is in agreement with the described role of ezrin as a major regulator linking the plasma membrane to the cortical actin while allowing the rapid remodeling of both structures[47,61,62].

The membrane-to-cortex architecture is tightly tuned in dynamic processes such as membrane protrusion formation during cell migration or membrane fusion in myoblasts[21,47,48,63]. Using AFM, we identified that BIN1 regulates the apparent static tether force. The absence of BIN1 would contribute to a disorganization of the $PI(4,5)P_2$/ezrin interface at the plasma membrane and consequently, to a loose membrane-to-cortex attachment and a decrease in the formation of filopodia. Indeed, the long membrane tails in migrating cells that we observed upon BIN1 knock-down would reflect a loosening of membrane-cortex attachments. Once filopodia formation is initiated, its growth rate (or retraction) is determined by the difference between actin polymerization speed and retrograde flow[64]. This retrograde flow arises from the friction between filopodia and lamellipodia since these structures appear connected at the cell surface[65]. Thus, while a loose membrane-cortex would favor the filopodia initiation process[60], it might have an opposite effect on later stages, for instance, affecting the friction required for a force production via retrograde flow during filopodia growing and secondly, on the membrane-actin cohesion of the resulting filopodia structure. Since filopodia formation is a dynamic process, affecting any of these stages would likely impact filopodia density, as we observed upon *Bin1* knock-down. Interestingly, all the BIN1-mediated processes here identified (F-actin bundling) and partners (IRSp53, dynamin) are mechanisms previously identified to participate in the formation of membrane protrusion required for myoblast fusion in mammals and insects[20,34,22,66–69]. Accordingly, we confirmed that BIN1 is required for myoblast fusion in C2C12 myoblasts, as previously reported in cellulo[25] and in vivo[26,70,71]). BIN1 isoform 8 appears dispensable for muscle development but is required for muscle regeneration in adulthood[70] and the D151N or the ΔSH3 BIN1 mutants are associated with the autosomal recessive form of centronuclear myopathies[24,43].

In conclusion, we discovered an unexpected role for BIN1 in the formation of filopodia, adding complexity in the diversity of the molecular mechanisms leading to filopodia formation[72]. Our study suggests a dual function of BIN1 both as a scaffold to recruit proteins, including ezrin, at the cell cortex and as an actin-bundling protein. We found a tandem role of IRSp53 and BIN1 in filopodia formation. The association of IRSp53 with antagonistic BAR domain proteins was reported to participate in filopodia formation[72,73] or to assist endocytosis[52,74] in different cell types and organisms. Why proteins with opposite curvature domains are required to enable distinct remodeling events at the plasma membrane is yet an open question but suggests that the spatio-temporal regulation of these events is a complex molecular process.

## Methods
### Reagents
Natural and synthetic phospholipids, including POPS, POPC, POPE, brain total lipid extract, brain L-α-phosphatidylinositol-4,5-bisphosphate, and

TopFluor-TMR-PI(4,5)P$_2$ are from Avanti Polar Lipids. OG-DHPE and β-casein from bovine milk (>98% pure) were from Sigma-Aldrich. Alexa Fluor 647 and 488 Maleimide labeling kits are from Invitrogen. Culture-Inserts 2 Well for self-insertion were purchased from ibidi.

The following antibodies were used in this study: monoclonal mouse antibody to BIN1 (clone C99D against exon 17) from Millipore, polyclonal rabbit antibody to IRSp53 (BAIAP2) from Atlas Antibodies (HPA023310), HRP-conjugated beta-actin monoclonal antibody from Proteintech (HRP-60008), ezrin antibody from M. Arpin laboratory[75], monoclonal rabbit antibody to phospho-Ezrin (Thr567)/Radixin (Thr564)/Moesin (Thr558) Cell Signaling (Cat. 3726). Atto390 phalloidin was from Sigma.

## Constructs
pGEX-Sumo vector coding for GST-BIN1 isoform 8 was obtained by Gibson assembly from ref. 28. pGEX vectors coding for GST-BIN1 isoform 8 ΔSH3 and D151N were obtained as in ref. 28. 6xHis-Sumo ezrin and T567D vectors were obtained as in ref. 8. pEGFP-BIN1 D151N, pEGFP-BIN1 ΔSH3 were obtained as in ref. 24. pEGFP-BIN1 isoform 8 was obtained from P. De Camilli (Yale University, New Haven). pEYFP-BIN1 isoform 1 and pEGFP-Amphiphysin1 were obtained from L. Johannes (Institut Curie UMR144, Paris). pEGFP-dynamin2 was a gift from Sandra Schmid (Addgene plasmid # 34686). mCherry-ezrin WT and T567 was obtained from ref. 76 after cloning into pENTR1A Gateway entry vector (Invitrogen) and recombined into mCherry-C1. full-length human IRSp53 (UniProt, no. Q9UQB8; Homo sapiens) was subcloned into the pGEX-6P-1 vector (Cytiva)[53]. All the plasmids used in this study (Table 1) were sequenced.

## Cell culture
HeLa cells (CCL2 from ATCC) were cultured in DMEM medium (Gibco BRL) supplemented with 10% fetal bovine serum, 100 U/ml penicillin/streptomycin, and 2 mM glutamine. C2C12 mouse myoblasts (ATCC CRL-1772) were grown in DMEM/Ham's F-12 (1:1) supplemented with 10% fetal bovine serum. To induce differentiation, the growth medium was replaced with differentiation medium consisting of DMEM/Ham's F-12 supplemented with 2% fetal calf serum (Hyclone/Perbio Sciences, Brebieres, France). All cells were tested mycoplasma free.

## Co-immunoprecipitation, western blot experiments, and proteomic assays
To test the interaction between Amphiphysin isoforms and endogenous actin or ERM proteins, HeLa cells (CCL2 from ATCC) were transfected for 24 h with either GFP, full-length GFP-BIN1 isoform 8, GFP-BIN1 isoform 1 and GFP-Amphiphysin1 using x-tremGENE9 (Roche). Cells were then trypsinized, washed once in PBS, and incubated on ice for 60 min in a lysis buffer: 25 mM Tris pH 7.5, 50–100 or 200 mM NaCl, and 0.1% NP40. Cells were then centrifuged 10 min at $10,000 \times g$ to collect the supernatant. Extracts were processed for co-immunoprecipitation using GFPTrap beads (Chromotek) (20 μl GFP-Trap beads per condition) for 3 h at 4 °C in lysis buffer. Beads were washed four times in lysis buffer and then processed for western blotting.

To test the interaction of BIN1 and its mutated variants, C2C12 myoblasts were transfected for 24 h with either GFP, full-length GFP-BIN1 isoform 8 WT, ΔSH3 and D151N mutants using JetPEI (Ozyme). Co-immunoprecipitation assay was performed as detailed above.

For western-blot experiments, cell lysis was performed in 25 mM Tris pH 7.5, 50 mM NaCl, 0.1% NP40, and a protease inhibitor cocktail (Sigma). The following primary antibodies were used: rabbit anti-ezrin (from M. Arpin laboratory[75]; 1:1000), mouse anti-actin (Sigma; 1:1000), mouse anti-BIN1 (clone C99D from Millipore; 1:1000), phosphor-ezrin. Secondary Horseradish Peroxidase (HRP)-coupled antibodies were from Jackson Laboratories.

For proteomic analysis, HeLa cells were transiently transfected with either GFP or GFP-BIN1 using X-tremeGENE 9 (Sigma-Aldrich), according to the manufacturer's instructions. HeLa cells, with high transfection efficiency, were used because we previously developed and set up proteomic analysis using this cell line and to ensure that cell type specificities would not bias the obtained BIN1 interactors. Cell lysis and co-immunoprecipitation using GFP-Trap (Chromotek) was performed as described above. Mass spectrometry analysis was performed at the Proteomic platform of Institut Jacques Monod (Paris, France). Positive hits binding to GFP-BIN1 were selected relatively to their corresponding Mascot score.

## Short interfering RNA (shRNA)
shRNA constructs were engineered on a pSIREN retroviral vector (Clontech). To deplete the endogenous expression of BIN1, the oligonucleotide 5′-GATCCG**CCTGATATCAAGTCGCGCATT**TTCAAGA-GA**AATGCGCGACTTGATATCAGG**CTTTTTTTACGCGTG-3′ was inserted into pSIREN. Bold letters correspond to exons 5/6's junction of mouse *Bin1*. As a control, we used the oligonucleotide 5′-GTTGCGCCCGCGAATGATATATAATGttcaagagaCATTATA-TATCATTCGCGGGCGCAAC-3′ sequence against luciferase.

HEK293T cells expressing shRNA *Bin1* or shRNA luciferase (CTRL) were cultured, and cell-free supernatants containing retrovirus were harvested. Two sequential transductions of C2C12 myoblasts were performed by adding 2 mL of filtered supernatant (0.45 mm PES sterile syringe filter) within a time window of 6 h between transductions. Double transduced cells were kept for 72 h in culture. After this time, cells were kept under puromycin at 1 μg/ml for 72 h before performing the experiments. For rescue experiments, cells were subsequently seeded overnight at 50% confluency on glass-coverslips and transfected for 24 h with Lifeact-mCherry and either GFP, GFP-BIN1 iso 8, GFP-BIN1 ΔSH3 or GFP-BIN1 D151N mutants using JetOPTIMUS (Ozyme). The cDNA of mouse *Bin1* contains three mismatches with the human forms of GFP-BIN1 iso8 and mutated variants. Resistance of human forms of BIN1 iso8 and mutants against the mouse shRNA *Bin1* was confirmed by fluorescence microscopy.

For each cycle of double transient transduction, shRNA efficiency was determined by western blotting to ensure a complete depletion of the BIN1 protein. Under these experimental conditions, we did not observe significant cellular mortality due to puromycin selection, suggesting that the entire population of cells should be infected by the shRNA.

To ensure that defects in cell proliferation and motility in the shRNA *Bin1* cell line, shRNA BIN1 C2C12 were seeded at higher density (i.e., at the onset of differentiation).

## RNA interference (siRNA)
The siRNA used in this study to IRSp53 was ON-TARGET plus siIRSp53 (BAIAP2) mouse (Horizon Discovery, Cat# J-046696-11)[77]. The siRNA sequence targeting luciferase (CGUACGCGGAAUACUUCGA) was used as a control and was obtained from Sigma. siRNA delivery was performed using Lipofectamine RNAiMAX, according to the manufacturer's instructions.

## Time-lapse fluorescence microscopy
C2C12 cells were seeded on FluoroDish (WPI, France) cell culture dishes overnight. Cells were then transfected for 12 h with either GFP-BIN1, D151N or ΔSH3 mutants and Lifeact-mCherry; GFP or GFP-BIN1 and its mutated variants and mCherry-ezrin using JetPEI (Ozyme) following the manufacturer's instructions. Live-cell imaging was performed on a Spinning disk microscope based on a CSU-X1 Yokogawa head mounted on an inverted Ti-E Nikon microscope equipped with a motorized XY Stage. Images were acquired through a 60x objective NA 1.4 Plan-Apo objective with a Photometrics Coolsnap HQ2 CCD camera. Optical sectioning was performed using a piezo stage (Mad City Lab). A dual Roper/Errol laser lounge equipped with 491 and 561 nm laser diodes (50 mW each) and coupled to the spinning disk head through a single fiber was used. Multi-dimensional acquisitions were performed in streaming mode using Metamorph 7 software. Images were collected every second (500 ms exposure) during 60 s.

**Table 1 | Summary of the plasmids used in this study**

| Designation | Protein species | Backbone | Source or reference | Additional information |
|---|---|---|---|---|
| pEGFP-BIN1-iso8 (plasmid) | Human | pEGFP-C1 | Lee et al. Science[26] | P. De Camilli Yale University New Haven USA |
| pEGFP-BIN1-D151N (plasmid) | Human | pEGFP-C1 | Nicot et al. Nat Genet[24] | J. Laporte IGBMC Illkirch, France |
| pEGFP-BIN1- ΔSH3 (plasmid) | Human | pEGFP-C1 | Nicot et al. Nat Genet[24] | J. Laporte IGBMC Illkirch, France |
| pEGFP-amphiphysin-1 (plasmid) | Human | pEGFP-C2 | | L. Johannes UMR144 Institut Curie, Paris, France |
| pEGFP-BIN1-iso1 (plasmid) | Human | pEYFP-C1 | | J. Laporte IGBMC Illkirch, France |
| pEGFP-dynamin2 (plasmid) | Rat | pEGFP-N1 | | Sandra Schmid (Addgene plasmid # 34686 |
| Lifeact-mCherry (plasmid) | | pIRES | Miserey-Lenkei et al.[83] | G. Montagnac INSERM U1279 Villejuif, France |
| mCherry-Ezrin (Gateway System) (plasmid) | Human | pmCherry-C1 | This study | M. Arpin laboratory. Institut Curie, Paris, France |
| mCherry-Ezrin-T567D (Gateway System) (plasmid) | Human | pmCherry-C1 | This study | M. Arpin laboratory. Institut Curie, Paris, France |
| IRSp53-mCherry (plasmid) | Human | pmCherry-N1 | Tsai et al. Science Advances[53] | F. Tsai UMR168 Institut Curie, Paris, France |
| pGEX-BIN1 (plasmid) | Human | pGEX6P1 | Picas et al. Nat Comm[28] | |
| 6xHis-Sumo-Ezrin (plasmid) | Human | pET28-N-His-SUMO | Tsai et al. eLife[8] | F. Tsai UMR168 Institut Curie, Paris, France |
| 6xHis-Sumo-Ezrin -T567D (plasmid) | Human | pET28-N-His-SUMO | Tsai et al. eLife[8] | F. Tsai UMR168 Institut Curie, Paris, France |
| pGEX-IRSp53 (plasmid) | Human | pGEX6P1 | Tsai et al. Science Advances[53] | F. Tsai UMR168 Institut Curie, Paris, France |
| pSIREN-ShRNA Luciferase (CTRL plasmid) | | pSIREN-RetroQ | This study | |
| pSIREN-ShRNA Bin1 (plasmid) | Mouse | pSIREN-RetroQ | This study | J. Laporte IGBMC Illkirch, France |

### Immunofluorescence microscopy

Fixed cells were obtained after incubation with 3.2% paraformaldehyde (PFA) for 10 min at room temperature, washed with PBS, incubated with PBS-0.1 M NH$_4$Cl for 5 min and then washed with PBS. Finally, cells were permeabilized in 0.1% Triton X-100 for 3 min and blocked with 1% BSA during 10 min. Fixed cells were mounted using mowiol mounting agent and visualized using a Leica DMRA and a CoolSnapHQ2 camera, 100× objective NA 1.25 oil Ph 3 CS (HCX PL APO) and analyzed with the Metamorph software. 3D stacks were acquired and deconvoluted to build a projection on one plane using Image J.

For myotube imaging, images were acquired on a Zeiss LSM880 Airyscan confocal microscope (MRI facility, Montpellier). Excitations sources used were: 405 nm diode laser, an Argon laser for 488 nm and 514 nm and a Helium/Neon laser for 633 nm. Acquisitions were performed on a 63 ×ì/1.4 objective. Multidimensional acquisitions were performed via an Airyscan detector (32-channel GaAsP photomultiplier tube (PMT) array detector). Images are presented as a z-projection of all planes.

**Tether pulling experiments.** Tether-pulling experiments were performed on a JPK Nanowizard III mounted on an inverted Zeiss wide-field microscope. Olympus Biolevers ($k = 6$ mN·m$^{-1}$) were cleaned in acetone for 5 min and then plasma-cleaned for 10 min. Then, cantilevers were soaked briefly in 0.1 M of NaHCO$_3$ (pH 9.0), air dried and immersed in 0.01% poly-L-lysin overnight at 4 °C in a humid chamber. Before the measurements, cantilevers were rinsed three times in PBS and mounted on the AFM cantilever holder. The cantilever spring constant was determined by the thermal noise method, as detailed in Schillers et al.[78]. For the measurement, cells seeded for 24 h on FluoroDish cell culture dishes were kept on DMEM-F12 medium at 37 °C and not used longer than 1 h for data acquisition. Static tether force measurements were performed by retracting the cantilever for 6 µm at a speed of 10 µm·s$^{-1}$, and the position was kept constant for 30 s. Resulting force–time curves were analyzed using the JPK analysis software. Tether force values represent the average value obtained from tether pulling maps performed over the whole apical membrane of adherent C2C12 cells after applying $3 \times 3$ grids of 10 µm$^2$.

The static tether force, $f_0$, depends on the bending stiffness of the membrane ($\kappa$), the in-plane membrane tension ($\sigma$), and the energy density of the membrane-to-cortex attachments ($W_0$)[49]:

$$f_0 = 2\pi\left(2\left(\sigma + W_0\right)\kappa\right)^{1/2}$$

### Protein purification and fluorescent labeling

Recombinant human full-length BIN1 isoform 8, Amphiphysin 1, ezrin wild-type and ezrin T567D were expressed in Rosetta 2 bacteria and purified by affinity chromatography using glutathione Sepharose 4B beads[8,28]. Recombinant proteins were labeled by conjugation with either Alexa Fluor 488 or 647 following maleimide chemistry (Invitrogen)[8,28].

Recombinant human full-length IRSp53 (Uniprot #Q9UQB8) was purified by affinity purification with GSTrap FF (Cytiva) and labeled with Alexa Fluor 488[53].

Muscle actin was purified from rabbit muscle and isolated in monomeric form in G-buffer (5 mM Tris-Cl-, pH 7.8, 0.1 mM CaCl2, 0.2 mM ATP, 1 mM DTT, 0.01% NaN3)[79].

### Lipid bilayer experiments

Supported lipid bilayers were prepared as described in ref. 80. The lipid mixture consisted of: 60% POPC, 20% POPE, 10% POPS and 5% PI(4,5)P$_2$. Fluorescent TopFluor-TMR-PI(4,5)P$_2$ was added to 0.1%. Experiments were performed by injecting 15 µL of buffer (10 mM Tris, pH 7.4, 100 mM NaCl and 0.5 mg·ml$^{-1}$ of casein). Supported lipid bilayers were imaged on a Zeiss LSM880 Airyscan confocal microscope (MRI facility, Montpellier). Excitations sources used were: Argon laser for 488 nm and 514 nm and a Helium/Neon laser for 633 nm. Acquisitions were performed on a 63×/1.4 objective.

### GUV preparation and observation

The lipid mixture used contains total brain extract supplemented with 5 mol% PI(4,5)P$_2$ at 0.5 mg/mL in chloroform. If needed, the lipid mixture is further supplemented with 0.5 mol% BODIPY TR ceramide or 0.5 mol% OG-DHPE.

GUVs were prepared by the polyvinyl alcohol (PVA) gel-assisted formation method[81]. Briefly, a PVA gel solution (5%, w/w, dissolved in

280 mM sucrose and 20 mM Tris, pH 7.5) warmed up to 50 °C was spread on clean coverslips (20 mm × 20 mm). The coverslips were cleaned with ethanol and then ddH$_2$O twice. The PVA-coated coverslips were incubated at 50 °C for 30 min. Then, around 5 µl of the lipid mixture was spread on the PVA-coated coverslips, followed by placing them under vacuum at room temperature for 30 min. The coverslips were then placed in a petri dish and around 500 µl of the inner buffer was pipetted on the top of the coverslips. The inner buffer contains 50 mM NaCl, 20 mM sucrose and 20 mM Tris-HCl pH 7.5. The coverslips were kept at room temperature for at least 45 min, allowing GUVs to grow. Once done, we gently "ticked" the bottom of the petri dish to detach GUVs from the PVA gel. The GUVs were collected using a 1 ml pipette tip with its tip cut to prevent breaking the GUVs.

For all experiments, coverslips were passivated with a β-casein solution at a concentration of 5 g.L$^{-1}$ for at least 5 min at room temperature. Experimental chambers were assembled by placing a silicon open chamber on a coverslip.

GUVs were first incubated with IRSp53 in the outer buffer (60 mM NaCl and 20 mM Tris-HCl pH 7.5) for at least 15 min at room temperature before adding BIN1 into the GUV-IRSp53 mixture. The final GUV-protein mixture was then incubated for at least 15 min at room temperature before observation. Samples were observed using a Nikon Eclipse Ti microscope equipped with Yokogawa CSU-X1 spinning disk confocal head, 100× CFI Plan Apo VC objective (Nikon) and a sCMOS camera Prime 95B (Photometrics).

### F-actin bundling assay

Actin (1 µM, non-labeled) was polymerized 1 h in a buffer containing 100 mM KCl, 1 mM MgCl$_2$, 0.2 mM EGTA, 0.2 mM ATP, 10 mM DTT, 1 mM DABCO, 5 mM Tris pH 7.5 and 0.01% NaN3 in the presence of BIN1 at different concentrations. Then, 5 µl of the protein mixtures was diluted 20 times (i.e., 50 nM of filamentous actin in the presence of BIN1 or BIN1 mutants) in the same buffer supplemented with 0.3% methylcellulose and 660 nM of Alexa Fluor 546-phalloidin. Samples were observed using TIRF microscopy (Eclipse Ti inverted microscope, 100× TIRF objectives, Quantum 512SC camera).

### Image processing and analysis

Protein binding was quantified by measuring the mean gray value of still confocal images of either AlexaFluor 488 ezrin, WT or T567D, in the absence of BIN1 or Amphiphysin 1. The obtained average intensity was then used to estimate the fold increase in the binding of ezrin, WT or T567D, in the presence of 0.1 µM of BIN1 or Amphiphysin 1 tagged with AlexaFluor 647. Each data set was performed using the same supported lipid bilayer preparation and confocal parameters were kept constant between experiments and samples. Mean gray values were measured once the steady-state of protein binding was reached, which was estimated to be ≥600 s. Mean gray values were measured using Image J[82].

To obtain actin fluorescence intensities on a filament or bundles, we manually defined a ROI, a 6 pixel-width line perpendicularly to the filament or bundle. We then obtained the intensity profile of the line in which the x-axis of the profile is the length of the line and the y-axis is the averaged pixel intensity along the width of the line. The actin intensity was the maximum intensity value in the intensity profile.

Filopodia density, morphology and dynamics were analyzed by a custom written program allowing the semi-automatic tracking the Life-actin signal of time-lapse movies using Image J.

### Statistics and reproducibility

Results are shown as a mean ± standard deviation. Unless stated otherwise, average values represent at least three technical replicates in the case of in vitro experiments and biological replicates in the case of cellular studies. Statistical significance was assessed by Welch's $t$ test, unless stated otherwise.

## Data availability

All data generated or analyzed during this study are included in the manuscript and supporting files; source data files have been provided as Supplementary Data. Unprocessed blots are presented in Supplementary Fig. 7. Mass spectrometry-based proteomics data is provided as Supplementary Data 2.

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

## Acknowledgements

The authors thank J. Viaud for kindly providing the PH-PLCd-Alexa647 probe. We thank Y. Senju (Okayama University, Japan) and Pekka Lappa-lainen (University of Helsinki, Finland) for IRSp53 purification and C. Le Clainche (Institute for Integrative Biology of the Cell, Gif-sur-Yvette, France) for actin purification. We also thank B. Cowling for the design of *Bin1* shRNA constructs. We also thank C. Cazevieille for assistance in TEM and J. Mateos-Langerak in SIM OMX imaging. The authors thank P. Sens for scientific discussions. The authors acknowledge the Nikon Imaging Center at Institut Curie-CNRS, the imaging facility MRI, the PICT-IBiSA, members of the national infrastructure France-BioImaging infrastructure supported by the French National Research Agency (ANR10-INBS-04). L.P. acknowl-edges the ATIP-Avenir program (AO-2016) for financial support. This project was also supported by grants from the Agence Nationale de la Recherche (ANR) (ANR-13-BSV2-0004-01), the Labex Cell(n)Scale (N° ANR-10- LBX-0038) part of the IDEX PSL (N° ANR-10-IDEX-0001-02 PSL), the Association Française contre les Myopathies (15352), the CNRS, the INSERM, the Uni-versity of Montpellier, University of Strasbourg, the College de France, and the Institut Curie.

## Author contributions

Conceptualization of the study: L.P., B.G., C.G.-R. and S.M. Performed experiments and data analysis: L.P., S.M., F.C., C.A.-A., H.B., F.-C.-T., J.P., F.R., P.M., S.B., A-S.N, and S.M. Supervision: L.P., J.L., P.B., B.G., C.G.-R., and S.M. Writing: L.P., C.G.-R. and S.M. with inputs from all authors.

## Competing interests

The authors declare no competing interests.
