## [Peer Review File · Communications Biology]

Reviewers' comments:

Reviewer #1 (Remarks to the Author):

This manuscript by Picas et al. identified two BIN1 binding partners IRSp53 and ezrin, and studied the functions of these proteins in regulating actin bundling and filopodia formation in muscle cells. Overall, the manuscript is well written and easy to follow, but the authors will have to provide more convincing data to fully illustrate the involvement of BIN1 in orchestrating molecular platform for promoting filopodia formation in cells.

Major points:

- 1) The authors performed shRNA to knockdown BIN1 and demonstrated its induced defects in filopodia formation, actin bundling and cell-cell junctions formation, etc. Replenishment of shRNA-resistant form of wild-type BIN1 and these two BIN1 mutants (D151N, Δ SH3 mutant) after knockdown, and to examine whether any of these phenotypes can be rescued in cells are important.
- 2) The authors provided some evidence that the SH3-domain of BIN1 mediates the interaction with ezrin, but it is unclear how BIN1 interacts with IRSp53?
- 3) The authors mentioned that BIN1 mutation is commonly related to congenital and adult centronuclear myopathies in human. I was wondering whether they could test any of these disease mutations in this study, to better illustrate the importance of BIN1 regulated actin-membrane interactions in IRSp53-dependent filopodia formation.

Minor points:

- 1) Fig. 3G is mentioned in the manuscript but is missing. Please double check that.
- 2) The detailed description on force measurement by AFM should be moved to the Methods.

Reviewer #2 (Remarks to the Author):

The authors identify a muscle-specific isoform of amphiphysin, BIN1-isoform8 ("BIN1"), as a regulator of myoblast fusion, by regulating filopodia formation and membrane-cortex attachment. ERM proteins, including ezrin, are identified as novel interaction partners of BIN1 and, the previously established interactions between BIN1 and F-actin, PI(4,5)P2 and IRSp53 are explored further, with IRSp53 depletion blocking the induction of filopodia by BIN1. A model is proposed where BIN1 contributes to filopodia formation (after initiation), by a contribution of F-actin bundling and recruitment of filopodial actin regulators. This contributes to a growing body of evidence that proteins which bind to positively curved membrane nonetheless promote formation of negatively curved membrane structures such as filopodia.

By identifying a role for BIN1 in promoting filopodial formation, and suggesting possible molecular mechanisms involved, this paper makes a useful contribution to general understanding of the range of molecules involved in filopodial formation, since most of the work exploring the role of BAR proteins in filopodial formation has focussed on I-BAR proteins such as IRSp53, which bind to negatively curved membrane. Importantly, the authors show that the promotion of filopodia has a function consequence

for myoblast fusion. Since this isoform of amphiphysin is specific to muscle cells, some of the specific interactions involved may not apply to other cell types, but it seems likely that the proposed mechanisms will be important more broadly.

The main argument, based on a combination of biophysical and cellular experiments, is generally convincing and data are mostly well described and presented, though a few key gaps should be addressed and in a number of places there are minor issues with text or data presentation which should be corrected.

Major comments

1. Though the importance of BIN1 for myoblast filopodia formation is clearly shown by the change in filopodia density, the evidence that BIN1 (or BIN1 + IRSp53/ezrin) is present at filopodia during formation appears limited to single examples of images/videos, so the authors should quantify the prevalence of BIN1 (or BIN1 + IRSp53/ezrin) at filopodia, either in fixed images or ideally in live images and state their numbers of experiments as this was not clear. The authors show that BIN1 interacts with IRSp53 in the proteomic screen and that knockdown of IRSp53 blocks the induction of filopodia by BIN1. However, the evidence presented for colocalisation between BIN1 and IRSp53 in cellular filopodia is a single filopodium, in which the BIN1 and IRSp53 signals barely overlap. Clearly both proteins localise to similar membrane structures, both in GUVs and cells, but to argue that “BIN1 associates with IRSp53 on membrane protrusions” there needs to be some quantification of the colocalisation. This could be showing more examples of more convincing colocalisation in filopodia, counting what proportion of filopodia have BIN1/IRSp53/both/neither etc., or some other way of showing a direct interaction in cellular filopodia. Another example is Fig 2F: The authors should quantify the actin bundling in the mutants, since the images alone are hard to interpret. In Fig S2C, a single low magnification image is shown as evidence of the absence of co-localisation between ezrin and BIN1 Δ SH3. It is very hard to judge if this is correct from this. Please show some higher magnification examples of FLS/filopodia. Quantification of the prevalence of how many FLS have BIN1 Δ SH3 or wt and ezrin would be helpful.

2. There are places in the text that are contradictory e.g. an “effect on filopodia density was observed between ezrin and the BIN1 Δ SH3 mutant”. However, in Fig 4E, the combination of ezrin and BIN1 Δ SH3 is significantly higher than either alone, and not significantly different from ezrin + BIN1 wt, which contradicts the model that the BIN1 SH3 domain is necessary for ezrin interaction. Please re-read carefully and resolve contradictions.

3. FLS lifetime cannot be established or compared when a large fraction, possibly a majority, of FLS exceed the maximum movie length (Fig. 2I). Either exclude this analysis or perform properly. Timestamps need including on all movies and montages. For several experiments the following explanation is given: “(500 msec exposure, during 60s. Total time acquisition = 120s)”. I don’t understand this sentence. The methods section seems to say that the total time is 60 s, but some of the figures show video timestamps or filopodia lifetimes up to 120 s. Please give more clarity and detail of the image acquisition.

4. Comparison of BIN1 iso1 and iso8 needs fuller description and discussion. Note that BIN1 iso1 does increase filopodia density significantly, by half as much as BIN1 iso8 (Fig 1A), and that it seems to localise

to protrusions (Fig 1A). Also, since BIN1 iso1 also bundles F-actin in vitro (albeit at a higher starting concentration), the authors could discuss why the two isoforms have different effects in cells.

Minor comments

5. The experiments relating to dynamin2 are unconvincing and not necessary for the overall argument (as BIN1 interaction with dynamin has previously been shown) so should be removed.

6. Sometimes it unclear which actin structures are of interest e.g. changes in nomenclature (“FLS” vs “filopodia”) so improve consistency for clarity.

7. Ensure images and figures are visible in paper copies. The text and panels are in places tiny such that it is impossible to communicate findings effectively. Labels are not always clear and consistently presented. Similarly good colour choices are not always used e.g. Fig 3A: It is hard to see IRSp53 in magenta with red actin present (and avoid red and green altogether), especially in top (low mag) image. In ROI image, show merge without actin to clarify colocalisation. Blue and yellow annotations are required in figures 4 and 5.

8. Fig 3A-3C appear to be not mentioned in the text. References to Fig 3D-3G seem duplicated; those in the first paragraph of section “IRSp53 is required for BIN1-mediated filopodia formation” would appear to instead refer to Fig 3A-3D. The text relating to Fig 3D talks about endogenous BIN1 iso8 staining but Fig 3A (presumably the relevant panel) shows GFP-BIN1 instead.

9. Discussion: It would be good to explore the unexpected result that, while BIN1 KD prevents phospho-ezrin enrichment to filopodia, it increases the abundance of phospho-ezrin. The authors say that loose membrane-cortex attachment would explain a decrease in filopodia formation? Why is this? Since filopodia involve the plasma membrane being pulled away from the cortex, it might instead be expected that this loosening would help filopodia formation. This is supported by the increase in “long membrane tails”.

Reviewers' comments:

Reviewer #1 (Remarks to the Author):

This manuscript by Picas et al. identified two BIN1 binding partners IRSp53 and ezrin, and studied the functions of these proteins in regulating actin bundling and filopodia formation in muscle cells. Overall, the manuscript is well written and easy to follow, but the authors will have to provide more convincing data to fully illustrate the involvement of BIN1 in orchestrating molecular platform for promoting filopodia formation in cells.

We thank the reviewer for his/her positive comments and suggestions to strengthen the involvement of BIN1 in filopodia formation.

Major points:

1) The authors performed shRNA to knockdown BIN1 and demonstrated its induced defects in filopodia formation, actin bundling and cell-cell junctions formation, etc. Replenishment of shRNA-resistant form of wild-type BIN1 and these two BIN1 mutants (D151N, Δ SH3 mutant) after knockdown, and to examine whether any of these phenotypes can be rescued in cells are important.

We agree with the reviewer that rescue experiments are important to confirm the phenotype of BIN1 expression in filopodia formation. Accordingly, we have included these experiments on the methods section (page 28) and manuscript page 8 and Fig. 2J, which summarize four independent replicates. The mouse shRNA *Bin1* used in this study targets exon 5 and 6 of the cDNA of mouse *Bin1* and contains three mismatches with the human forms of GFP-BIN1 iso8 and mutated variants used in this study. We confirmed by fluorescence microscopy that the transfection of hBIN1 iso8 and mutants were indeed resistant to the mouse shRNA *Bin1*. We found that expression of BIN1 iso8 on shRNA *Bin1* knock-down C2C12 reestablished filopodia density, whereas this was not the case for the two BIN1 mutants, D151N and Δ SH3 mutant, in agreement with our observations in Fig. 2G-I indicating that both the N-terminal and C-terminal region of BIN1 are likely to contribute in BIN1-mediated filopodia formation.

2) The authors provided some evidence that the SH3-domain of BIN1 mediates the interaction with ezrin, but it is unclear how BIN1 interacts with IRSp53?

The point raised by the reviewer is a good one. Accordingly, we have performed co-immunoprecipitation experiments (GFP-Trap) by expressing GFP-BIN1 iso8 and the D151N and Δ SH3 mutant to detect by WB their association with IRSp53. The results, which are representative of three independent replicates, are now presented on page 10 and Fig. 3C. Our results show that both wild-type BIN1 and D151N associate with IRSp53 *in cellulo*, whereas this is not the case of the Δ SH3 mutant. These results suggest that the SH3 domain of BIN1 mediates the interaction with different BIN1 partners involved in filopodia formation, like ezrin and IRSp53.

3) The authors mentioned that BIN1 mutation is commonly related to congenital and adult

centronuclear myopathies in human. I was wondering whether they could test any of these disease mutations in this study, to better illustrate the importance of BIN1 regulated actin-membrane interactions in IRSp53-dependent filopodia formation.

We acknowledge the reviewer for his/her comment, and we agree that it might be very interesting to determine the role of IRSp53 in patients suffering from CNMs and presenting BIN1 mutations, especially in the C-terminal domain. However, at present, this type of test is, in addition to being technically challenging, out of the scope of this work, which was the identification and cellular characterization of new BIN1 functions and the co-factors involved in this process. Nevertheless, we do not exclude investigating the physiopathology of BIN1/IRSp53 interaction in the future, possibly by developing the appropriate mouse models.

Minor points:

1) Fig. 3G is mentioned in the manuscript but is missing. Please double check that.

We thank the reviewer for bringing up this typo, which is now corrected.

2) The detailed description on force measurement by AFM should be moved to the Methods.

Following the reviewer's suggestion, this part has now been moved to the methods section on page 28.

Reviewer #2 (Remarks to the Author):

The authors identify a muscle-specific isoform of amphiphysin, BIN1-isoform8 ("BIN1"), as a regulator of myoblast fusion, by regulating filopodia formation and membrane-cortex attachment. ERM proteins, including ezrin, are identified as novel interaction partners of BIN1 and, the previously established interactions between BIN1 and F-actin, PI(4,5)P2 and IRSp53 are explored further, with IRSp53 depletion blocking the induction of filopodia by BIN1. A model is proposed where BIN1 contributes to filopodia formation (after initiation), by a contribution of F-actin bundling and recruitment of filopodial actin regulators. This contributes to a growing body of evidence that proteins which bind to positively curved membrane nonetheless promote formation of negatively curved membrane structures such as filopodia.

By identifying a role for BIN1 in promoting filopodial formation, and suggesting possible molecular mechanisms involved, this paper makes a useful contribution to general understanding of the range of molecules involved in filopodial formation, since most of the work exploring the role of BAR proteins in filopodial formation has focussed on I-BAR proteins such as IRSp53, which bind to negatively curved membrane. Importantly, the authors show that the promotion of filopodia has a function consequence for myoblast fusion. Since this isoform of amphiphysin is specific to muscle cells, some of the specific interactions involved may not apply to other cell types, but it seems likely that the proposed mechanisms will be important more broadly.

The main argument, based on a combination of biophysical and cellular experiments, is generally convincing and data are mostly well described and presented, though a few key gaps should be addressed and in a number of places there are minor issues with text or data presentation which should be corrected.

We thank the referee for his/her encouraging comments and are pleased to see that he/she finds our work of importance. Below, we address the points raised by the reviewer.

Major comments

1. Though the importance of BIN1 for myoblast filopodia formation is clearly shown by the change in filopodia density, the evidence that BIN1 (or BIN1 + IRSp53/ezrin) is present at filopodia during formation appears limited to single examples of images/videos, so the authors should quantify the prevalence of BIN1 (or BIN1 + IRSp53/ezrin) at filopodia, either in fixed images or ideally in live images and state their numbers of experiments as this was not clear. The authors show that BIN1 interacts with IRSp53 in the proteomic screen and that knockdown of IRSp53 blocks the induction of filopodia by BIN1. However, the evidence presented for colocalisation between BIN1 and IRSp53 in cellular filopodia is a single filopodium, in which the BIN1 and IRSp53 signals barely overlap. Clearly both proteins localise to similar membrane structures, both in GUVs and cells, but to argue that "BIN1 associates with IRSp53 on membrane protrusions" there needs to be some quantification of the colocalisation. This could be showing more examples of more convincing colocalisation in filopodia, counting what proportion of filopodia have BIN1/IRSp53/both/neither etc., or some other way of showing a direct interaction in cellular filopodia. Another example is Fig 2F: The authors

should quantify the actin bundling in the mutants, since the images alone are hard to interpret. In Fig S2C, a single low magnification image is shown as evidence of the absence of co-localisation between ezrin and BIN1 Δ SH3. It is very hard to judge if this is correct from this. Please show some higher magnification examples of FLS/filopodia. Quantification of the prevalence of how many FLS have BIN1 Δ SH3 or wt and ezrin would be helpful.

Following the reviewer's recommendation, we have included the corresponding quantifications in the revised version of the manuscript: We have now quantified the actin-bundling activity of recombinant BIN1 iso8 and the D151N and Δ SH3 mutants in Fig. 2F. We also include new data supporting the cellular association (in line with reviewer 1's comment, see above) and co-localization of BIN1 and IRSp53 at filopodia (Fig. 3A-C and page 10). We also provide new data through co-immunoprecipitation assays that IRSp53 associates with BIN1 iso8 and the D151N mutant *in cellulo* (Fig. 3C). Furthermore, we provide quantification of the % of filopodia showing a co-localization of BIN1 and IRSp53 (Fig. 3B) to support the representative images shown in Fig. 3A. The same quantification has also been performed with mCherry-ezrin to show the % of filopodia that display a co-localization between ezrin and BIN1 iso8 or BIN1 mutants (Fig. 4E). Finally, we have included magnified images to show the absence of co-localization between ezrin and the BIN1 Δ SH3 mutant in Fig. S2C.

2. There are places in the text that are contradictory e.g. an "effect on filopodia density was observed between ezrin and the BIN1 Δ SH3 mutant". However, in Fig 4E, the combination of ezrin and BIN1 Δ SH3 is significantly higher than either alone, and not significantly different from ezrin + BIN1 wt, which contradicts the model that the BIN1 SH3 domain is necessary for ezrin interaction. Please re-read carefully and resolve contradictions.

We thank the reviewer for pointing out this relevant mistake. Indeed, in an earlier version of our manuscript, Fig. 4E (now Fig. 4F in the revised version) did not display the condition "BIN1 iso8". We then decided to include it to compare the different phenotypes better. Unfortunately, when doing this operation, the new graph that was generated (as it was displayed in the previous Fig. 4E) did not match the former data, which is why there was a contradiction between the text and the graph. Following the reviewer's remark, we have now amended this mistake.

3. FLS lifetime cannot be established or compared when a large fraction, possibly a majority, of FLS exceed the maximum movie length (Fig. 2I). Either exclude this analysis or perform properly. Timestamps need including on all movies and montages. For several experiments the following explanation is given: "(500 msec exposure, during 60s. Total time acquisition = 120s)". I don't understand this sentence. The methods section seems to say that the total time is 60 s, but some of the figures show video timestamps or filopodia lifetimes up to 120 s. Please give more clarity and detail of the image acquisition.

We agree with the reviewer that the description of the time-lapse experiments might need to be clarified. Indeed, we applied 500 msec exposure during 60 s for the GFP and mCherry channels sequentially, giving a final total time of 120s (2 x 60s). We have removed the term "during 60 s", as it might be misleading, and kept "total time = 120 s" in the final version of the manuscript.

Endogenous filopodia have a lifetime of 79 to 142s (<https://doi.org/10.1016/j.semcdb.2009.11.008>), depending on the cell type. Indeed, this is what we observe in our control (empty GFP) condition, where more than 75% of filopodia show a lifetime \leq 120s. This trend is significantly modified upon expression of GFP-BIN1 iso8 and mutants, and this is what we wanted to show with our experimental setup. We agree with the reviewer that increasing the time acquisition would provide an absolute value of the average filopodia lifetime. However, this would also imply increasing cell phototoxicity due to more prolonged laser exposure time, which must be considered when designing an experiment. Therefore, we reasoned that 120 s would be a good compromise to show a significant effect in filopodia dynamics without compromising cell viability.

4. Comparison of BIN1 iso1 and iso8 needs fuller description and discussion. Note that BIN1 iso1 does increase filopodia density significantly, by half as much as BIN1 iso8 (Fig 1A), and that it seems to localise to protrusions (Fig 1A). Also, since BIN1 iso1 also bundles F-actin in vitro (albeit at a higher starting concentration), the authors could discuss why the two isoforms have different effects in cells.

The reviewer is right in pointing out that other BIN1/amphiphysin family members promote filopodia formation, although to a lesser extent, and that this could better explained. Indeed, *in vitro*, BIN1 iso8 already displays an actin-bundling ability at lower concentrations than the neuronal isoform (BIN1 iso 1, see Dräger et al., 2017), which might explain the differences between the two isoforms in promoting filopodia formation *in cellulo*. However, other factors could also contribute to differences in the filopodia density phenotype observed. For instance, the selective targeting of BIN1 iso8 to the plasma membrane through its PI(4,5)P2-binding motif (Lee et al., 2002), which is not present in BIN1 iso1, might enhance its function on PI(4,5)P2-mediated processes such as actin remodeling (Senju et al., 2017; Senju and Lappalainen, 2019). Also, the contribution of other cellular factors, such as IRSp53, could not be excluded, and it would be interesting to address further its contribution to recruiting different BIN1/amphiphysin isoforms at negatively-curved membranes based on the homology of their SH3 domains (Prokic et al., 2014).

The above explanation has now been included in the discussion section, page 21.

Minor comments

5. The experiments relating to dynamin2 are unconvincing and not necessary for the overall argument (as BIN1 interaction with dynamin has previously been shown) so should be removed.

We understand the reviewer's concern about the experiments on dynamin2, and we agree that these experiments do not constitute the core of our work. However, we decided to include them to validate the identification of new BIN1 partners involved in filopodia formation since two relevant works have recently identified dyanmin2 as a central player involved in actin-protrusion formation during myoblast fusion (<https://doi.org/10.1038/s41556-020-0519-7> and <https://doi.org/10.1083/jcb.201809161>). In light of the reviewer's comment, we have removed all the experiments relating dynamin2 from the main manuscript and moved them to supplementary

information (Fig. S4), as we believe these observations might still be interesting for the scientific community.

6. Sometimes it unclear which actin structures are of interest e.g. changes in nomenclature (“FLS” vs “filopodia”) so improve consistency for clarity.

We agree that the nomenclature needed to be more consistent. Accordingly, we have changed all the nomenclature to “filopodia.”

7. Ensure images and figures are visible in paper copies. The text and panels are in places tiny such that it is impossible to communicate findings effectively. Labels are not always clear and consistently presented. Similarly good colour choices are not always used e.g. Fig 3A: It is hard to see IRSp53 in magenta with red actin present (and avoid red and green altogether), especially in top (low mag) image. In ROI image, show merge without actin to clarify colocalisation. Blue and yellow annotations are required in figures 4 and 5.

We acknowledge the reviewer for bringing up all these improvements. Accordingly, we have increased the panels' text font and improved the labels' consistency. We have also changed the color display of Fig. 3A and SFig. S4 to avoid red and green altogether. The co-localization between GFP-BIN1 iso8 and IRSp53 is now shown in the merged image and through a profile analysis in Fig. 3A. Finally, we have specified the description of blue and yellow displays in the figure caption of Fig. 4 and 5.

8. Fig 3A-3C appear to be not mentioned in the text. References to Fig 3D-3G seem duplicated; those in the first paragraph of section “IRSp53 is required for BIN1-mediated filopodia formation” would appear to instead refer to Fig 3A-3D. The text relating to Fig 3D talks about endogenous BIN1 iso8 staining but Fig 3A (presumably the relevant panel) shows GFP-BIN1 instead.

Following the reviewer's remark, Fig. 3A-C is now mentioned on page 10. In addition, all panels in Fig. 3 are now appropriately indicated in the manuscript. Finally, we amended the sentence on page 10 to "we confirmed by immunofluorescence that endogenous IRSp53 colocalize with GFP-BIN iso8 at filopodia on C2C12 myoblasts (Fig. 3A)."

9. Discussion: It would be good to explore the unexpected result that, while BIN1 KD prevents phospho-ezrin enrichment to filopodia, it increases the abundance of phospho-ezrin. The authors say that loose membrane-cortex attachment would explain a decrease in filopodia formation? Why is this? Since filopodia involve the plasma membrane being pulled away from the cortex, it might instead be expected that this loosening would help filopodia formation. This is supported by the increase in “long membrane tails”.

We agree with the reviewer that the increase in phospho-ezrin that we observed upon Bin1 KD is an unexpected yet interesting observation that, at the moment, we cannot fully explain, including what might be the specific cellular localization and function of this ezrin hyper-phosphorylation.

The reviewer is right, pointing out that the effect of BIN1 in the membrane-cortex mechanics needs to be better discussed, and, accordingly, we have reviewed this in the discussion section (page 22). Indeed, the reviewer is correct that a loose membrane cortex should favor filopodia formation since pushing a filopodium out of a membrane under tension requires force. This issue has been nicely addressed recently by the Danuser lab (Welf et al., 2020), showing that local depletion of ezrin from the plasma membrane favors filopodia initiation, as we point out on pages 21-22 of our manuscript. Since BIN1 requires IRSp53 for its localization at negatively-curved membranes, the contribution of BIN1 in filopodia formation likely takes place after the protrusion is initiated, as we argue in the discussion section. However, once the structure is initiated, filopodia growth rate (or retraction) is determined by the difference between actin polymerization speed and retrograde flow, as explained in doi: 10.1002/cm.21130. This retrograde flow arises from the friction between filopodia and lamellipodia since these structures appear connected at the cell surface (doi: 10.1083/jcb.200210174.). Thus, while a loose membrane-cortex would favor the filopodia initiation process, it might have an opposite effect on later stages, for instance, affecting the friction required for a force production via retrograde flow during filopodia growing and secondly, on the membrane-actin cohesion of the resulting filopodia structure. Since filopodia formation is a dynamic process, affecting any of these stages would likely impact filopodia density, as we observe upon BIN1 knock-down.

Reviewers' comments:

Reviewer #1 (Remarks to the Author):

The authors have performed substantial new experiments, and the new results have cleared all my concerns. I have no more comments.

Reviewer #2 (Remarks to the Author):

I can recognize that the authors have made some effort to respond to the reviews. The sub panels are extremely small which makes reading and evaluating this manuscript very difficult. Two points of my review are not adequately handled:

(Point 1) Quantification of IRSp53 / BIN1 localization. Nothing is visible in 3A and the inset shows they are both dotted all over, calling this co-localization is misleading. Area of overlap might be an alternative in this situation. % of filopodia with each protein and the quantified effect of the mutation would be better. The ezrin data is more convincing showing the time-lapse and quantified effect of the mutants in 4D/E.

(Point 3) Claim of a change in filopodia lifetime quantified in 2I. The statistics done on this are not valid if the imaging time is less than filopodia lifetime. If you cannot image the process you are seeking to study for technical reasons then you have no data. The explanation is incorrect too, in a situation of phototoxicity then there are several ways to adjust like a different imaging method, more sensitive camera, brighter fluorophore, lower exposure time or images need to be acquired less frequently for a longer time period. Showing the fraction of filopodia with a lifetime less than a certain value might be a workaround but is not ideal. As presented with the artificial ceiling plus statistics is incorrect.

Reviewers' comments:

Reviewer #2 (Remarks to the Author):

I can recognize that the authors have made some effort to respond to the reviews. The sub panels are extremely small which makes reading and evaluating this manuscript very difficult.

We thank the reviewer for his/her comments and suggestions to improve the manuscript.

Following his/her remark, we have increased the policy of the graphs in the revised version of the manuscript.

Two points of my review are not adequately handled:

(Point 1) Quantification of IRSp53 / BIN1 localization. Nothing is visible in 3A and the inset shows they are both dotted all over, calling this co-localization is misleading. Area of overlap might be an alternative in this situation. % of filopodia with each protein and the quantified effect of the mutation would be better. The ezrin data is more convincing showing the time-lapse and quantified effect of the mutants in 4D/ E.

Following the reviewers' remark, we have performed new experiments using GFP fusion versions of BIN1 and BIN1 mutants together with IRSp53-mCherry to avoid the dotted effect upon endogenous labeling of IRSp53 by immunofluorescence. The data corresponding to the colocalization of GFP-BIN1 iso8 and endogenous IRSp53 is now moved to Fig. S3.

We present in page 12 a new figure with the colocalization of IRSp53-mCherry and GFP-BIN1 iso8 (Fig. 3A) or BIN1 mutants (Fig. S3), along with a quantification of the overlap of the signal of BIN1 and BIN1 mutants with IRSp53-mCherry (Fig. 3B). This is, we manually quantified the % of filopodia positive for an overlapping of between mCherry/GFP signals versus filopodia only positive for IRSp53. Unfortunately, we did not manage to quantify the precise area of overlap between the two signals at the filopodia level.

(Point 3) Claim of a change in filopodia lifetime quantified in 2I. The statistics done on this are not valid if the imaging time is less than filopodia lifetime. If you cannot image the process you are seeking to study for technical reasons then you have no data. The explanation is incorrect too, in a situation of phototoxicity then there are several ways to adjust like a different imaging method, more sensitive camera, brighter fluorophore, lower exposure time or images need to be acquired less frequently for a longer time period. Showing the fraction of filopodia with a lifetime less than a certain value might be a workaround but is not ideal. As presented with the artificial ceiling plus statistics is incorrect.

We thank the reviewer for his/her suggestion. We propose a new analysis that we believe is more correct after the reviewer's remark in page 8 and Fig. 2I-J. We show

in Fig. 2I the % of dynamic filopodia versus the % of static filopodia, which is defined as the filopodia that do not collapse in a period of time ≥ 120 s. We have performed a chi-square test for discrete statistics between proportions for the different conditions. We also present the mean lifetime of dynamic filopodia (i.e., filopodia that do not die in a period ≤ 120 s) in Fig. 2J.